# NOMAD: Lifelong Trajectory Planning via Non-Parametric Bayesian Memory-Adaptive Diffusion Experts

**Yixian Chen** [* 1]  **Rufan Bai** [* 2]  **Jiangbin Zheng** [3]  **Yimin Wang** [4]  **Tiantian Chen** [5]  **Wei Wang** [6]  **Yuhuan Lu** [7]

## Abstract

Autonomous vehicles operating in open-world environments must continually adapt to rare long-tail scenarios while preserving previously acquired driving skills. However, existing trajectory planning approaches struggle with this stability–plasticity trade-off, as they rely on static models or rigid rule-based controllers that cannot robustly handle evolving and complex traffic dynamics. Against this background, we propose **NOMAD**, a lifelong trajectory planning framework that integrates non-parametric Bayesian memory with diffusion-based trajectory generation, enabling continuous adaptation to long-tail scenarios without catastrophic forgetting. Our method maps growing scene contexts to a dynamically growing set of discrete memory clusters, which guide a conditional diffusion model to function as a mixture of experts specialized for diverse driving behaviors. To retain past knowledge during incremental learning, we introduce a generative replay mechanism that synthesizes pseudo-experiences from previously learned memory clusters. Extensive closed-loop evaluations on the nuPlan benchmark demonstrate that our approach achieves state-of-the-art performance on long-tail scenarios, improving the interPlan score by **9.4%** over the strongest baseline, while maintaining competitive performance on regular driving benchmarks. Moreover, our method exhibits robust continual learning capability, achieving the highest average closed-loop score with positive backward transfer when adapting to sequentially introduced long-tail scenarios.

## 1. Introduction

Trajectory planning serves as the decision-making core of autonomous driving systems, acting as the critical bridge that translates perception and prediction data into actionable control commands. While accurate forecasting of surrounding agents is necessary, the ultimate goal of an autonomous vehicle (AV) is closed-loop planning (Karchanachari et al., 2024; Hallgarten et al., 2024), that is, the ability to dynamically interact with traffic participants, correct deviations in real-time, and ensure safety and stability during execution. Consequently, developing robust planners that can operate reliably in complex, real-world environments remains a challenge in the field.

Current approaches to trajectory planning are predominantly categorized into rule-based and learning-based methods. Rule-based planners (Treiber et al., 2000; Dauner et al., 2023) rely on explicit, hand-crafted heuristics to determine vehicle behavior. These methods offer high interpretability and safety within well-defined operational domains but are inherently fragile as they struggle to generalize to complex, interactive scenarios that fall outside their predefined logical boundaries. Conversely, learning-based planners (Cheng et al., 2024b;a; Yang et al., 2024a; Zheng et al., 2025; 2026) leverage large-scale driving datasets to learn policies that imitate expert human driving. Models such as PLUTO (Cheng et al., 2024a) and Diffusion Planner (Zheng et al., 2025) have demonstrated the ability to capture diverse and multimodal behaviors. However, these data-driven approaches often remain largely opaque and are susceptible to causal confusion, distribution shifts, and a lack of safety guarantees when encountering out-of-distribution data.

A critical limitation shared by both paradigms is the lack of generalization to the long tail of rare driving events. Once deployed, current planning models are typically static. They

---

[*]Equal contribution  [1]School of Design, Foshan University, Foshan, China [2]Key Laboratory of Computer Network and Information Integration, Southeast University, Nanjing, China [3]School of Software, Northwestern Polytechnical University, Xi'an, China [4]School of Computer Science and Engineering, Sun Yat-Sen University, Guangzhou, China [5]Cho Chun Shik Graduate School of Mobility, Korea Advanced Institute of Science and Technology, Daejeon, South Korea [6]Engineering Research Centre of Applied Technology on Machine Translation and Artificial Intelligence, Macao Polytechnic University, Macao, China [7]Faculty of Applied Sciences, Macao Polytechnic University, Macao, China. Correspondence to: Wei Wang <weiwang@mpu.edu.mo>, Yuhuan Lu <yc17462@connect.um.edu.mo>.

*Proceedings of the $43^{rd}$ International Conference on Machine Learning*, Seoul, South Korea. PMLR 306, 2026. Copyright 2026 by the author(s).

lack the capacity to continuously acquire new skills from streaming data without suffering from catastrophic forgetting, where learning new behaviors degrades the performance of previously mastered tasks. This inability to adapt is particularly harmful when an AV encounters unseen scenarios such as different cities, evolving traffic norms, or accident corner cases. Therefore, equipping trajectory planning systems with the capability for continual learning, allowing them to accumulate knowledge over a lifespan and adapt to new domains without extensive retraining, is a key challenge for achieving higher-level autonomy.

While recent research has begun to address this adaptation gap, existing solutions are either unable to learn continuously or are limited by the low capabilities of rule-based controllers. While promising, such knowledge-driven approaches often face latency bottlenecks due to heavy LLM inference and rely heavily on the retrieval quality of stored experiences, which may not suffice for rapid, reactive closed-loop control. Our approach addresses this by mapping the continuous trajectory generation process onto a discrete knowledge structure. We assume that the infinite stream of driving scenarios can be decomposed into distinct latent clusters, where each cluster represents a specific type of driving skill or environmental context.

We present **NOMAD**, a **NO**n-parametric Bayesian **M**emory **A**daptive **D**iffusion planner. Instead of using heuristic clustering, we employ a Dirichlet Process Mixture Model (DPMM) to automatically discover and maintain the discrete structure of driving scenarios from the continuous stream of perception data. This memory module dynamically expands to accommodate new long-tail cases without requiring predefined cluster numbers. To generate high-quality trajectories, we employ a mixture of conditional diffusion planner that serves as the trajectory planner. The diffusion model is conditioned on the semantic context retrieved from the discrete memory, effectively allowing it to specialize its generation strategy for different scenario clusters. Furthermore, we address the stability-plasticity dilemma in lifelong learning through a generative replay mechanism. Since our planner is a generative model, we utilize it to synthesize pseudo-data from previously learned memory clusters. These synthetic samples are mixed with new data during updates, ensuring that the model retains mastery of common scenarios while adapting to new long-tail cases. Finally, to guarantee safety during execution, we incorporate classifier guidance into the diffusion inference process, ensuring that the generated trajectories satisfy kinematic and safety constraints.

We evaluate our framework on the large-scale nuPlan benchmark, focusing specifically on hard long-tail scenarios. Our method demonstrates the ability to incrementally learn from a stream of rare cases while maintaining high performance on learned tasks. The main contributions of this paper are

summarized as follows:

- We propose NOMAD, a lifelong learning trajectory planner that integrates non-parametric Bayesian inference with diffusion-based planning. This architecture enables continuous adaptation to open-world scenarios while maintaining the high expressiveness of generative models.

- We introduce a mechanism that maps continuous scene embeddings to a dynamic discrete memory using a Dirichlet Process Mixture Model. This allows the system to automatically identify and record novel long-tail scenarios without manual supervision.

- Extensive closed-loop evaluations on the nuPlan benchmark validate the effectiveness of our approach. Our method improves the interPlan long-tail score by **9.4%** over the strongest baseline, with consistent gains across challenging scenarios. Importantly, these improvements do not degrade performance on regular driving benchmarks. Moreover, our method exhibits robust continual learning capability, with the average closed-loop score progressively improving as new long-tail scenarios are sequentially introduced, accompanied by positive backward transfer.

## 2. Related Work

### 2.1. Autonomous Vehicle Trajectory Planning

#### 2.1.1. RULE-BASED PLANNING

Rule-based planners rely on explicit heuristics to ensure safety and interpretability. The Intelligent Driver Model (IDM) (Treiber et al., 2000) serves as a foundational car-following model but lacks lateral maneuvering capabilities. To address this, PDM-Closed (Dauner et al., 2023) extends IDM by simulating multiple trajectory proposals with varying speeds and lateral offsets, selecting the optimal path via a scoring function. While PDM-Closed achieved state-of-the-art on the nuPlan benchmark, it struggles to generalize to complex, long-tail scenarios due to its limited behavioral space.

#### 2.1.2. LEARNING-BASED PLANNING

Data-driven approaches have evolved to address the limitations of handcrafted rules.

**Imitation Learning (IL)**: Early methods like Urban Driver (Scheel et al., 2022) leveraged PointNet encoders and global reasoning to clone expert policies. Recent models include PlanTF (Cheng et al., 2024b), Pluto (Cheng et al., 2024a), and GameFormer (Huang et al., 2023), which integrated more advanced deep learning architectures to enhance planning generalization.

**Reinforcement Learning (RL)**: To overcome the distribution shift inherent in IL, CarPlanner (Zhang et al., 2025a) proposes a consistent auto-regressive framework trained via RL. It utilizes expert-guided rewards and a non-reactive transition model to generate multi-modal trajectories, outperforming rule-based baselines on large-scale benchmarks.

**Diffusion Models**: Generative diffusion models have emerged for modeling complex multi-modal distributions. Diffusion-ES (Yang et al., 2024a) integrates diffusion with evolutionary search to optimize black-box reward functions without gradients. To ensure real-time feasibility, DiffusionDrive (Liao et al., 2025) introduces a truncated diffusion policy initialized from anchor trajectories. Similarly, Diffusion Planner (Zheng et al., 2025) jointly models ego-planning and neighbor prediction using a transformer-based architecture, employing classifier guidance to align outputs with safety constraints.

**Knowledge-driven**: Large Language Models (LLMs) are increasingly used for semantic reasoning, like LLM-Assist (Sharan et al., 2023) and PlanAgent (Zheng et al., 2026). Recently, DAPlanner (Zhang et al., 2025b) utilizes a dual-agent framework in Frenet space, where one agent generates trajectories and another discriminates them to ensure safety.

### 2.1.3. HYBRID PLANNING

Hybrid architectures aim to combine the robustness of rules with the generalization of learning. HybridLLMPlanner (Hallgarten et al., 2024) pairs an LLM-based behavior selector with a rule-based motion planner to navigate rare scenarios, while SAH-Drive (Fan et al., 2025) utilizes a dual-timescale decision neuron to dynamically switch between a PDM-Closed planner for regular driving and a diffusion planner for long-tail cases.

### 2.2. Continual Learning for Autonomous Driving

Continual Learning (CL) addresses the "stability-plasticity" dilemma, enabling agents to learn sequentially from data streams without catastrophic forgetting (Wang et al., 2024). Classical CL strategies generally fall into three categories: replay-based methods (Lopez-Paz & Ranzato, 2017; Rebuffi et al., 2017) that store and retrain on past exemplars; regularization-based methods (Kirkpatrick et al., 2017; Li & Hoiem, 2017; Nguyen et al., 2018) that constrain weight updates to preserve prior knowledge; and architecture-based methods (Mallya & Lazebnik, 2018; Rosenfeld & Tsotsos, 2018) that dynamically expand model capacity for new tasks.

In the specific domain of trajectory planning, CL is essential for adapting to evolving driving scenarios of different traffic patterns or geographies. LSTOL (Yang et al., 2024b) investigates online learning frameworks that allow agents to learn in-situ without forgetting previously acquired detection and planning capabilities. To address domain shifts, such as transferring between cities with different traffic rules (e.g., left- vs. right-hand traffic), LoRD (Diehl et al., 2025) introduces a Low-Rank Residual Decoder. This approach freezes the base model and learns a lightweight residual module to adapt to Out-of-Distribution scenarios effectively. Most recently, LiloDriver (Yao et al., 2025) proposed a lifelong learning framework for long-tail scenarios. It integrates LLMs with a memory bank to retrieve and refine planning strategies for rare cases, enabling the system to continuously accumulate driving experience and adapt to unseen environments without extensive retraining.

## 3. Method

In this section, we present our framework designed for closed-loop trajectory planning in autonomous driving. Our approach addresses the challenge of adapting to long-tail scenarios through a lifelong learning paradigm. We first define the problem formulation. Subsequently, we present the overall architecture, which comprises three core components organized into two parts. The upper part contains the Hierarchical Scene Encoder and the Non-Parametric Bayesian Memory, while the lower part consists of the Mixture of Conditional Diffusion Experts Planner. Finally, we describe the generative replay mechanism used to mitigate catastrophic forgetting and the safety guidance mechanism applied during inference.

### 3.1. Problem Formulation

We consider an autonomous vehicle planning through a dynamic environment. At any time step $t$, the system receives an observation $O_t$. This observation is a composite data structure containing the high-definition map features denoted as $M$ and the state history of dynamic agents denoted as $A$. The map $M$ includes lane centerlines, road boundaries, and crosswalks. The agent data $A$ includes the position, velocity, and heading of surrounding vehicles and pedestrians over the past $T_h$ historical seconds.

The objective of the planner is to synthesize a future trajectory $\tau$ for the ego vehicle. We define $\tau$ as a sequence of $T_f$ waypoints, represented as $\tau = \{s_1, s_2, \ldots, s_{T_f}\}$, where each state $s_i$ comprises the 2D position $(x_i, y_i)$ and the heading angle $\theta_i$.

Unlike a standard supervised learning model trained on a fixed dataset, we address the lifelong learning setting where the data arrives in a continuous stream $\mathcal{S} = \{(O_t, \tau_t)\}_{t=1}^{\infty}$. The distribution of driving scenarios $P_t(O)$ is non-stationary. For example, the vehicle may initially operate in a city with regular grid traffic and later encounter a rural area with roundabouts and unpaved roads. The goal is to learn a

policy that maximizes the likelihood of expert trajectories across all seen environments up to time $t$, without storing the entire history of raw observations.

## 3.2. Architecture Overview

Our framework features a modular architecture and a disentangled training scheme, as illustrated in Figure 1. The overall workflow of NOMAD proceeds as follows. First, an alternating optimization scheme is adopted in the upper part. The scene encoder maps raw observations to compact latent embeddings. Following a fixed number of encoder training steps, the Non-Parametric Bayesian Memory applies a Dirichlet Process Mixture Model to the scene embeddings, identifying distinct driving scenario types. Subsequently, with the upper part parameters frozen, the diffusion experts planner in the lower part is trained conditioned on both the scene embedding and the cluster context. The training data comprises a mixture of current scenario samples and generative replay samples to mitigate catastrophic forgetting. The detailed training and inference algorithms are provided in Appendix A.1.

The disentangled training pipeline is motivated by the following principles:

- Noise isolation from the lower planner for stable learning. The diffusion model can generate substantial gradient noise in early stage, which would otherwise degrade the learned scene representations.

- Functional separation between knowledge preservation and trajectory planning to avoid conflicting objectives. Concretely, the upper part learns scenario-level representations and identifies cluster structures, whereas the lower part is dedicated to trajectory generation.

- Protection of the non-parametric memory for continual learning. Decoupling ensures DPMM operates over a stable embedding distribution, preserving cluster centers corresponding to previously learned knowledge.

Empirical and visual validation for the adoption of this training scheme is provided in Appendix A.2.

## 3.3. Hierarchical Scene Encoder

The Scene Encoder transforms complex environmental data into a fixed-length latent vector $z$. We adopt a hierarchical encoding structure similar to recent transformer-based planners (Yao et al., 2025). The input consists of vectorized map elements and surrounding agent tracks. We use MLP-Mixer blocks to extract local features from each map polygon and agent trajectory individually. These local features are then aggregated using a standard Multi-Head Self-Attention

(Vaswani et al., 2017) mechanism to capture global interactions. The final output is a latent scene embedding $z \in \mathbb{R}^d$ that encapsulates the semantic information of the driving environment. Finally, the scene encoder is updated by minimizing the KL divergence between the distribution of scene encoder output and the assigned cluster components from DPMM, denoted as $\mathcal{L}_{\mathbf{KL}}$.

## 3.4. Non-Parametric Bayesian Memory

A core challenge in lifelong autonomous driving is that the total number of driving scenarios is infinite and cannot be predefined. To address this, we employ a Non-Parametric Bayesian Memory. Unlike static clustering methods used in prior works, we utilize a Dirichlet Process Mixture Model (Lee et al., 2024) to organize the latent space.

We treat the scene embeddings $z$ as observations generated from a mixture of latent components. The DPMM allows the number of components, denoted as $K$, to grow with data. Let $G \sim \mathrm{DP}(\alpha, \mathcal{H})$ be a random probability measure drawn from a Dirichlet process with concentration parameter $\alpha$ and base distribution $\mathcal{H}$ parameterized by $\lambda$. Using the stick-breaking construction, the mixing proportions $\boldsymbol{\pi} = (\pi_1, \pi_2, \dots)$ are defined as:

$$\beta_k \sim \mathrm{Beta}(1, \alpha), \quad \pi_k = \beta_k \prod_{l=1}^{k-1}(1 - \beta_l) \qquad (1)$$

where $\pi_k$ is the probability of assigning an observation to component $k$. The generative process for each scene embedding $z_i$ is then:

$$\theta_k \sim \mathcal{H}(\lambda) \qquad (2)$$
$$v_i \sim \mathrm{Cat}(\boldsymbol{\pi}) \qquad (3)$$
$$z_i \sim F(\theta_{v_i}) \qquad (4)$$

Here, $v_i$ is a latent cluster assignment drawn from the categorical distribution over components, $\theta_{v_i}$ are the component-specific parameters drawn from the base distribution, and $F$ is an exponential family distribution parameterized by $\theta_{v_i}$.

To preserve the non-parametric nature despite truncation of $K$, we use memoized Variational Bayes (memoVB) (Hughes & Sudderth, 2013) to update the posterior of DPMM, which processes data in batches via additive sufficient statistics. We approximate the intractable posterior $p(\boldsymbol{v}, \boldsymbol{\theta}, \boldsymbol{\beta}|\mathbf{z})$ with a variational distribution $q(\boldsymbol{v}, \boldsymbol{\theta}, \boldsymbol{\beta})$ by maximizing the Evidence Lower Bound (ELBO). Upon encountering a long-tail scenario that poorly aligns with existing clusters, the memoVB automatically triggers a birth move to instantiate a new component. Conversely, if incorporating this scenario into an existing cluster yields an improvement in the overall ELBO, the memoVB performs a merge move. This mechanism allows the memory to expand

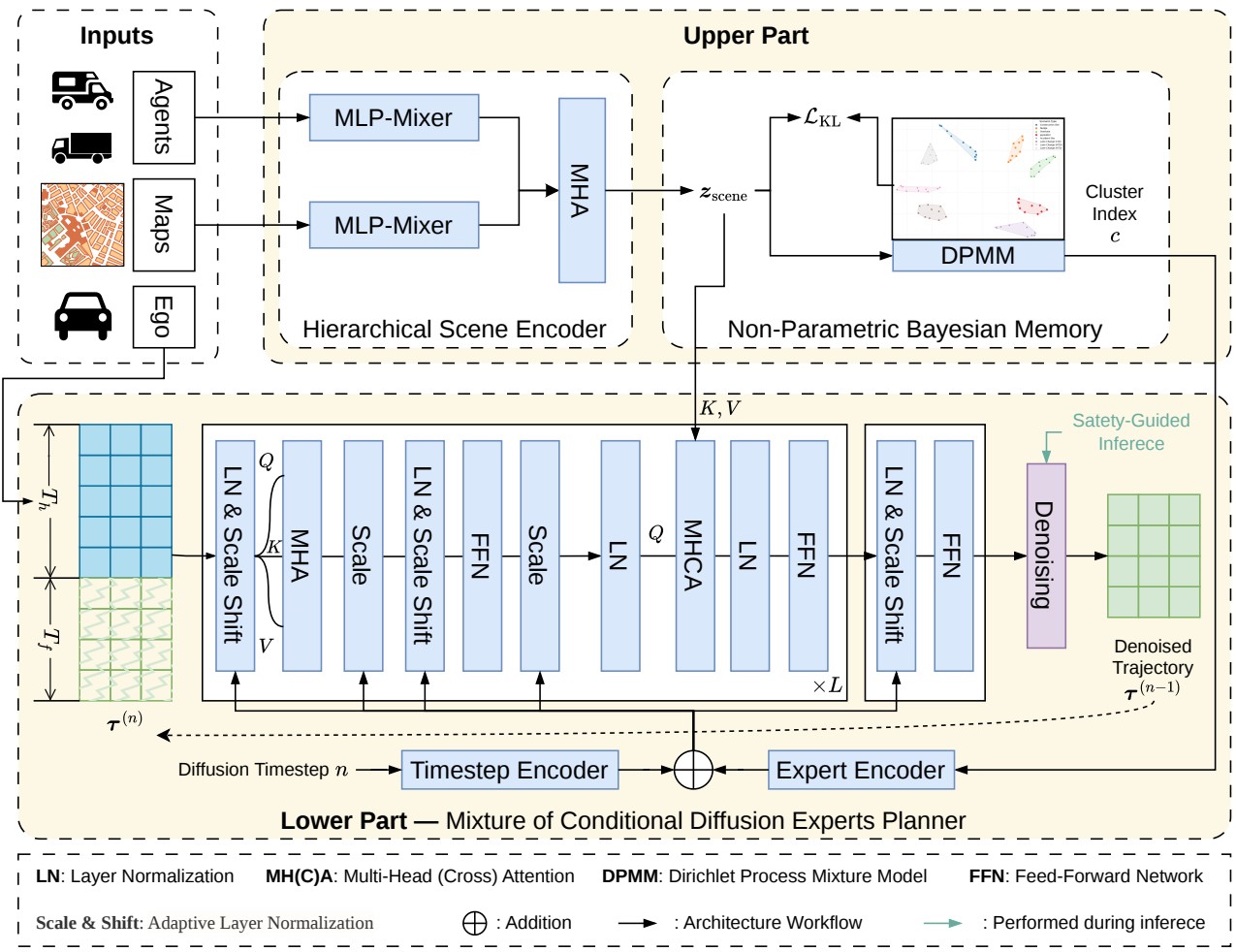

*Figure 1.* **The Overall architecture of NOMAD**.

adaptively as novel long-tail scenarios appear. The identified cluster assignment serves as a discrete context $c$ which guides the planner in the lower part. This approach ensures that distinct driving scenarios, such as unprotected left turns or merging into highways, are separated in the memory space. The variational inference of DPMM is detailed in Appendix B.

### 3.5. Mixture of Conditional Diffusion Experts Planner

We reformulate the planning module as a conditional trajectory generation process using diffusion models (Sohl-Dickstein et al., 2015; Ho et al., 2020). Unlike methods that output low-dimensional controller parameters (Yao et al., 2025), diffusion models can capture complex multi-modal trajectory distributions (Fan et al., 2025; Zheng et al., 2025).

We construct a conditional diffusion model parameterized by $\phi$, denoted as $\epsilon_\phi$. The model learns to denoise a Gaussian noise variable $\tau_N$ into a feasible trajectory $\tau_0$ through a reverse diffusion process:

$$p_\phi(\tau_{n-1}|\tau_n, z, c) = \mathcal{N}(\tau_{n-1}; \mu_\phi(\tau_n, n, z, c), \Sigma_\phi(\tau_n, n)) \tag{5}$$

The key innovation here is the conditioning on the memory context $c$. By feeding the cluster assignment $c$ from the Non-Parametric Bayesian Memory into the diffusion model, we effectively instantiate a mixture of diffusion experts planner, analogous to the manner in which a Gaussian mixture model (Reynolds et al., 2009) is composed of shared component distributions. The single diffusion network $\epsilon_\phi$ acts as a shared backbone, while $c$ modulates the generation process to act as a specialized expert for that specific scenario type.

The training objective is to minimize the noise prediction error:

$$\mathcal{L}_{\text{diff}} = \mathbb{E}_{\tau_0, \epsilon, n, z, c} \left[ \|\epsilon - \epsilon_\phi(\tau_n, n, z, c)\|^2 \right] \tag{6}$$

where $n$ is the diffusion time step, $\epsilon \sim \mathcal{N}(0, I)$, and $\tau_n$ is

the noisy trajectory at step $n$. The conditioning on $z$ captures instance-specific features while $c$ routes to scenario-specialized behavior. The architectural design of mixture of conditional diffusion experts planner is detailed in Appendix C.

### 3.6. Generative Replay for Lifelong Learning

Adapting the diffusion model to new clusters can lead to catastrophic forgetting of old driving skills. To solve this, we introduce a generative replay mechanism. This approach avoids storing raw historical data, which improves privacy and storage efficiency.

When the DPMM identifies a new cluster and the model updates its parameters, we use a frozen copy of the previous model, $\phi_{\text{old}}$, to generate synthetic trajectories for previously learned clusters. Specifically, we sample diverse scene embeddings $z$ from the fitted Gaussian components in the memory and pair them with their cluster indices $c$. The frozen model generates corresponding trajectories $\tau_{\text{gen}}$. These generated pairs $(z, c, \tau_{\text{gen}})$ are mixed with the data from the new task to form a replay dataset. The diffusion model is then trained on this combined dataset. This ensures that the distribution of the mixture of experts is maintained across all learned scenarios.

In Appendix H, we evaluate the effectiveness and fidelity of generative replay samples through a series of experiments.

### 3.7. Safety-Guided Inference

A major advantage of diffusion-based planning is the flexibility to enforce constraints during inference. While the memory module provides the prior for how to drive in a specific scenario, we must also ensure real-time safety and comfort. We employ classifier guidance to refine the generated trajectory (Dhariwal & Nichol, 2021).

We define a differentiable energy function $\mathcal{J}(\tau)$ that quantifies safety violations, such as target speed maintenance (Zheng et al., 2025). During the reverse sampling steps, we adjust the estimated mean of the trajectory distribution using the gradient of this energy function:

$$\hat{\mu} = \mu_\phi(\tau_n, n, z, c) - \gamma \Sigma_\phi(\tau_n, n) \nabla_{\tau_n} \mathcal{J}(\tau_n) \quad (7)$$

where $\gamma$ is a guidance scale parameter. This gradient step steers the generated trajectory towards safer regions in the state space without requiring retraining of the base model. This allows our planner to handle unexpected obstacles or tight dynamic constraints that may not have been fully captured in the training distribution. The specific energy functions used for safety and comfort are detailed in Appendix D.

## 4. Experiment

We evaluate **NOMAD** in closed-loop simulation to answer two core questions: (1) whether NOMAD achieves strong *standard planning performance* on both regular and long-tail scenarios, and (2) whether NOMAD can *continually acquire new long-tail skills* without degrading previously learned behaviors. All experiments are conducted on the nuPlan benchmark (Karnchanachari et al., 2024).

### 4.1. Datasets and Metrics

**Datasets.** We follow standard nuPlan evaluation protocols and report results on both regular and long-tail splits. Specifically, we evaluate on **Val14** and **Test14-Random** for common driving scenarios, and on **interPlan** (Hallgarten et al., 2024) and **Test14-Hard** for long-tail scenarios. For continual learning experiments, long-tail scenarios are sequentially introduced following the protocol in (Yao et al., 2025).

**Metrics.** Closed-loop score reactive (CLS-R) and closed-loop score non-reactive (CLS-NR) metrics, evaluated by nuPlan (Karnchanachari et al., 2024), are used. The evaluative dimension include safety, compliance, and driving quality of an autonomous driving system. For evaluation metrics of continual learning, refer to Appendix E.

### 4.2. Main Performance

**Standard Performance.** Tables 1 and 2 summarize the closed-loop performance of NOMAD against SOTA planners, where the baselines are introduced in Appendix F.

On the long-tail **interPlan** benchmark, NOMAD achieves the highest overall score, outperforming hybrid methods such as SAH-Drive and learning-based planners, including LiloDriver and Diffusion Planner, across most scenario categories. Notably, NOMAD shows consistent gains in challenging scenarios such as construction zones, accidents, overtaking, and high-density lane changes, indicating strong generalization beyond rule-based or single-mode behaviors.

On regular benchmarks (Val14 and Test14-Random), NOMAD matches or exceeds prior SOTA performance, demonstrating that improvements on long-tail scenarios do not come at the cost of degraded performance in common driving conditions. Compared to Diffusion Planner, NOMAD benefits from memory-guided specialization, while compared to SAH-Drive, NOMAD avoids heuristic switching and instead learns scenario-adaptive behaviors end-to-end.

**Continual Learning.** Figure 2 reports the continual learning performance of NOMAD under incremental long-tail scenario adaptation. Starting from a model trained on common driving scenarios, previously unseen long-tail scenarios are sequentially introduced.

| Planner | Type | interPlan | Constr. | Acc. | Jayw. | Nudge | Overt. | LTD | MTD | HTD |
|---|---|---|---|---|---|---|---|---|---|---|
| **SOTA** | | | | | | | | | | |
| PDM-Closed (CoRL 2023) | Rule | 42 | 18 | 0 | 48 | 74 | 9 | 62 | 62 | 62 |
| STR2 (ICRA 2025) | Learning | 46 | / | / | / | / | / | / | / | / |
| HybridLLMPlanner (IROS 2024) | Hybrid | 53 | 27 | 20 | 48 | **93** | 28 | 81 | 48 | **80** |
| Diffusion-ES (CVPR 2024) | Learning | 57 | 71 | 51 | 13 | 88 | 52 | 61 | 58 | 61 |
| PlanTF (ICRA 2024) | Learning | 33 | 9 | 0 | 33 | 49 | 9 | 50 | 40 | 73 |
| Pluto (arxiv 2024) | Learning | 48 | 54 | 9 | 56 | 82 | 17 | 47 | 47 | 68 |
| Diffusion Planner (ICLR 2025) | Learning | 24 | 17 | 0 | 7 | 70 | 15 | 41 | 22 | 17 |
| SAH-Drive (ICML 2025) | Hybrid | 64 | 72 | 44 | 47 | 80 | 78 | 64 | 63 | 63 |
| LiloDriver (arxiv 2025) | Learning | 56 | 60 | 41 | 45 | 82 | 73 | 65 | 60 | 64 |
| NOMAD (Ours) | Learning | **70** | **76** | **58** | **67** | 90 | **84** | **83** | **69** | 75 |
| | | +9.4% | +5.5% | +13.7% | +1.5% | -3.2% | +7.7% | +2.5% | +9.5% | -6.3% |
| **Suboptimal** | | | | | | | | | | |
| Urban Driver (CoRL 2022) | Learning | 4 | 0 | 0 | 0 | 0 | 0 | 0 | 29 | 0 |
| GameFormer (ICCV 2023) | Learning | 11 | 0 | 0 | 48 | 0 | 0 | 0 | 20 | 21 |
| DTPP (ICRA 2024) | Learning | 25 | 18 | 18 | 44 | 10 | 0 | 40 | 36 | 34 |
| IDM (Phys. Rev. E) | Rule | 31 | 0 | 0 | 66 | 0 | 0 | 61 | 61 | 61 |

*Table 1.* **Overall performance comparison between NOMAD and SOTA planners on the interPlan benchmark**. interPlan evaluates planning performance under eight long-tail driving scenarios, including construction zones, accident scenes, jaywalkers, nudging, overtaking, and lane-changing with low, medium, and high traffic densities. The best performance in each metric is highlighted in **bold**, while the second-best is underlined. The last row reports the relative improvement of NOMAD over the strongest baseline for each metric.

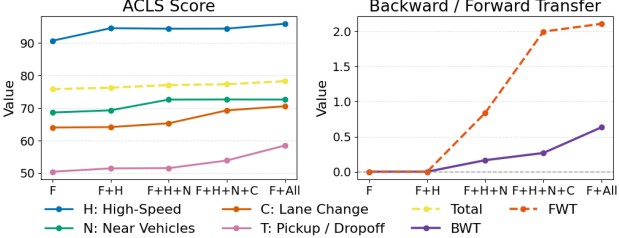

*Figure 2.* **Performance of memory-enhanced lifelong continual learning on Test14-Hard**. Starting from 10 common driving scenarios (**F**), we incrementally incorporate previously unseen long-tail scenarios into the memory module and evaluate performance across different long-tail settings, including **H**: high-magnitude speed, **N**: near multiple vehicles, **C**: lane changing, and **T**: traversing pickup/drop-off zones.

Across all stages, NOMAD exhibits a progressively increasing ACLS as new long-tail scenarios are introduced, with rapid performance gains following each incremental update. The positive BWT indicates that NOMAD effectively mitigates catastrophic forgetting, preserving previously learned driving behaviors. At the same time, NOMAD maintains competitive FWT, demonstrating sufficient plasticity to adapt to new scenarios.

These results suggest that NOMAD strikes a favorable balance between stability and adaptability in lifelong trajectory planning. For a more detail comparison with other continual learning strategies, refer to Appendix G.

### 4.3. Ablation Study

Table 3 presents ablation results to analyze the contribution of key components in NOMAD. Removing the non-parametric knowledge memory (**w/o Memory**) leads to a substantial drop in overall performance, particularly in rare scenarios such as accidents, jaywalking, and high-density lane changes. This confirms that explicitly modeling scenario-level structure is crucial for capturing long-tail driving behaviors. Without memory guidance, the planner tends to collapse to conservative strategies or fails to generalize beyond frequently observed patterns.

Replacing the hierarchical scene encoder (**w/o Encoder**) also results in consistent performance degradation across most scenario categories. This indicates that structured scene representations are essential for robust clustering and downstream trajectory generation. A weaker encoder produces less discriminative embeddings, which in turn degrades both memory assignment and expert specialization in the diffusion planner.

Finally, substituting the memory-conditioned diffusion model with a standard diffusion planner in SAH-Drive (**w/ Normal Diffusion**) reduces performance even when all other components are retained. This suggests that the mixture-of-experts formulation is critical for leveraging the discrete memory. Without scenario-conditioned modulation, a single diffusion model struggles to represent diverse and potentially conflicting driving behaviors, especially under

| Planner | Type | interPlan | Val14 (R) | Val14 (NR) | Test14-Random (R) | Test14-Random (NR) | Test14-Hard (R) | Test14-Hard (NR) |
|---|---|---|---|---|---|---|---|---|
| **SOTA** | | | | | | | | |
| PDM-Closed | Rule | 42 | 92 | 93 | 91 | **90** | 75 | 65 |
| STR2 | Learning | 46 | 93 | / | / | / | / | / |
| HybridLLMPlanner | Hybrid | 53 | / | / | / | / | / | / |
| Diffusion-ES | Learning | 57 | 92 | / | / | / | 77 | 77 |
| PlanTF | Learning | 33 | 77 | 84 | 80 | 85 | 61 | 69 |
| Pluto | Learning | 48 | 78 | 89 | 78 | 89 | 60 | 70 |
| Diffusion Planner | Learning | 24 | 83 | 90 | 83 | 89 | 69 | 75 |
| SAH-Drive | Hybrid | 64 | 90 | 89 | 87 | 86 | 83 | **78** |
| LiloDriver | Learning | 56 | 79 | 84 | 83 | 85 | 79 | 77 |
| CarPlanner | Learning | 60 | 88 | 86 | 91 | **90** | 80 | 72 |
| DAPlanner | Hybrid | 55 | 87 | 85 | 84 | 85 | 81 | 70 |
| NOMAD (Ours) | Learning | **70** | **95** | **94** | **93** | 88 | **85** | **78** |
| **Suboptimal** | | | | | | | | |
| UrbanDriver | Learning | 4 | 50 | 69 | 67 | 52 | 49 | 50 |
| GameFormer | Learning | 11 | 75 | 80 | 82 | 84 | 67 | 68 |
| DTPP | Learning | 25 | 73 | / | / | / | / | / |
| IDM | Rule | 31 | 77 | 75 | 74 | 70 | 62 | 56 |

*Table 2.* **Overall performance comparison between NOMAD and SOTA planners on different splits of the nuPlan benchmark.** Results are reported on interPlan (long-tail), Val14 (regular), Test14-Random (regular), and Test14-Hard (long-tail). The best performance in each metric is highlighted in **bold**, while the second-best is underlined.

| Variant | interPlan | Constr. | Acc. | Jayw. | Nudge | Overt. | LTD | MTD | HTD |
|---|---|---|---|---|---|---|---|---|---|
| Original | **70** | **76** | **58** | **67** | **90** | **84** | **83** | **69** | **75** |
| w/o Memory | 64(-8.57%) | 76(0.00%) | 49(-15.52%) | 54(-19.40%) | 83(-7.78%) | 80(-4.76%) | 67(-19.28%) | 69(0.00%) | 70(-6.67%) |
| w/o Encoder | 68(-2.86%) | 70(-7.89%) | 50(-13.79%) | 66(-1.49%) | 85(-5.56%) | 81(-3.57%) | 80(-3.61%) | 60(-13.04%) | 68(-9.33%) |
| w/ Normal Diffusion | 67(-4.29%) | 73(-3.95%) | 55(-5.17%) | 67(0.00%) | 90(0.00%) | 82(-2.38%) | 78(-6.02%) | 68(-1.45%) | 71(-5.33%) |

*Table 3.* **Ablation study on the memory design, scene encoder, and diffusion model.** The percentage following each score indicates the relative change with respect to the original configuration.

long-tail conditions.

Taken together, the ablation study demonstrates that the gains of NOMAD do not stem from a single component. Instead, strong performance arises from the synergy between structured scene encoding, non-parametric memory, and memory-guided diffusion-based trajectory generation.

### 4.4. Qualitative Results

**Long-tail Planning Scenario.** Figure 3 visualizes representative long-tail scenarios to qualitatively compare NOMAD with strong baselines. In the emergent accident scenario, NOMAD is the only method that proactively detects the blocked road segment and generates a safe re-routing trajectory. The w/o Memory variant and SAH-Drive continue to follow front vehicles, resulting in deadlock behaviors. This highlights the importance of memory-guided scenario awareness, enabling NOMAD to recall and apply appropriate behaviors for rare but safety-critical situations.

In the overtaking scenario, all compared methods are able to generate feasible trajectories. However, clear behavioral differences emerge. The w/o Memory variant produces overly conservative trajectories, hesitating to initiate overtaking despite sufficient free space. SAH-Drive generates more aggressive trajectories but occasionally violates traffic rules,

such as running a red light. In contrast, NOMAD generates trajectories that balance efficiency and compliance, completing overtaking maneuvers while respecting traffic signals. These results demonstrate that NOMAD not only improves success rates but also yields more socially compliant and context-aware behaviors.

**Visualization of Latent Space.** Given that the proposed memory leverages a DPMM for dynamic clustering of latent scene embeddings, we visualize the resulting clusters to demonstrate their correspondence to meaningful planning scenarios. Figure 4 is the t-SNE visualization of the interPlan scene embeddings generated by DPMM. It shows 8 long-tail scenario types forming clearly separated clusters, with quantitative cluster purity 100% against ground-truth labels. Semantically, each cluster corresponds to a distinct driving behavior (e.g., construction site, jaywalker, overtake).

In addition, we visualize the emergent clusters in Appendix J as continual learning progresses and the number of encountered scenarios continues to grow.

These qualitative results suggest that NOMAD benefits from explicitly disentangling scenario structure through non-parametric memory. Rather than relying on heuristic planner switching or post-hoc correction, NOMAD directly

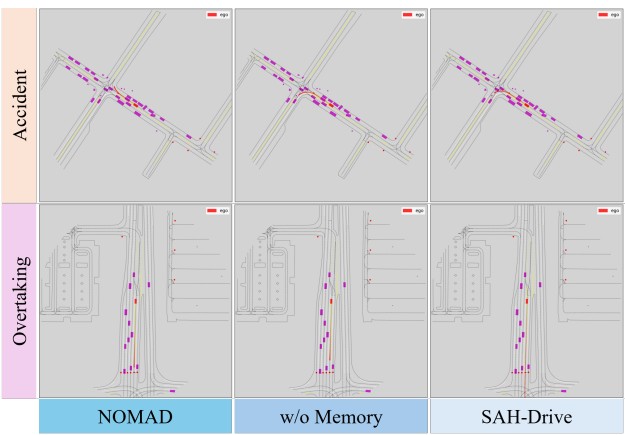

*Figure 3.* **Visualization of two long-tail scenarios: an emergent accident and an overtaking case**. The red rectangle denotes the ego vehicle. In the emergent accident scenario, only our NOMAD method detects the hazard and replans the route, whereas the w/o Memory variant and SAH-Drive continue to follow the front vehicles. In the overtaking scenario, all three methods generate feasible overtaking trajectories; however, the w/o Memory variant produces overly conservative behaviors, while SAH-Drive generates trajectories that violate traffic signals (running a red light).

conditions trajectory generation on learned scenario clusters, leading to more stable and interpretable behaviors in diverse long-tail settings.

| Method | interPlan (s) | Test14 (s) |
|---|---|---|
| PDM-Closed | $0.53 \pm 0.03$ | $0.60 \pm 0.05$ |
| Diffusion-ES | $5.85 \pm 0.11$ | $6.21 \pm 0.14$ |
| Diffusion Planner | $2.82 \pm 0.15$ | $2.73 \pm 0.24$ |
| SAH-Drive | $1.89 \pm 0.10$ | $1.72 \pm 0.14$ |
| NOMAD (Ours) | $1.46 \pm 0.08$ | $1.50 \pm 0.09$ |

*Table 4.* **Running time comparison on the interPlan and Test14 benchmarks**. We report the average wall-clock inference time per frame (mean $\pm$ standard deviation).

### 4.5. Runtime Analysis

Table 4 compares inference time across planners. NOMAD achieves real-time performance with an average inference time of approximately 1.5 seconds per frame, comparable to SAH-Drive and significantly faster than Diffusion-ES. This efficiency stems from a single diffusion backbone shared across experts and avoids expensive planner switching or large language model inference. Sensitivity analysis of model size is presented in Appendix I. Overall, NOMAD offers a favorable balance between performance and computational cost.

### 4.6. Robustness and Parameter Sensitivity Analysis

We further conduct experiments evaluating robustness to partial observations and sensitivity to the guidance scale parameter $\gamma$, with full details provided in Appendix K.

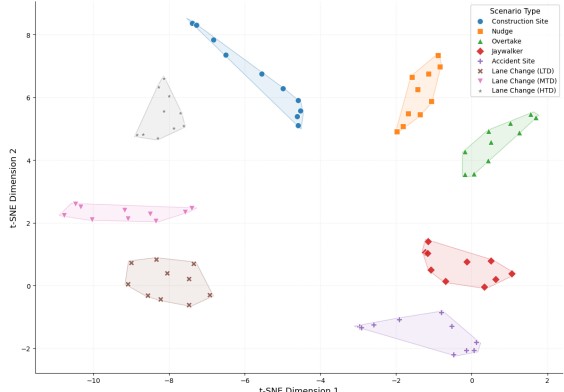

*Figure 4.* **t-SNE Visualization of interPlan Scene Embeddings Generated by DPMM**. 8 long-tail scenario types form clearly separated clusters, each corresponding to a distinct driving behavior (e.g., navigating a construction site, avoiding a jaywalker, overtaking).

## 5. Conclusion

In this work, we proposed **NOMAD**, a lifelong trajectory planning framework that integrates non-parametric Bayesian memory with diffusion-based trajectory generation. By mapping growing driving scenes to a dynamically expanding discrete memory and conditioning trajectory generation on scenario-specific contexts, NOMAD effectively bridges static planning and continual adaptation in open-world driving environments.

Extensive closed-loop evaluations on the nuPlan benchmark demonstrate that NOMAD achieves SOTA performance across both regular and long-tail scenarios. In particular, NOMAD improves the interPlan long-tail score by **9.4%** over the strongest baseline, while exhibiting progressively increasing average closed-loop performance as new long-tail scenarios are sequentially introduced, with positive backward transfer indicating minimal forgetting. Qualitative analyses and ablation studies further confirm that these gains arise from the synergy among hierarchical scene encoding, non-parametric memory, mixture of conditional diffusion experts planning, and generative replay.

**Limitations.** As illustrated in Appendix L, NOMAD still struggles in scenarios with weak semantic cues, such as sparse parking-lot environments. In these cases, limited scene structure makes reliable memory assignment and trajectory generation challenging. Addressing such scenarios may require richer scene representations or explicit spatial reasoning, which we leave for future work.

## Acknowledgements

This study is funded in part by the National Natural Science Foundation of China under Grant No. 62402103, in part by

the Natural Science Foundation of Jiangsu Province under Grant No. BK20241273, in part by the National Natural Science Foundation of China under Grant No. 62572120, and in part by the Natural Science Foundation of Jiangsu Province under Grant No. BK20230024.

## Impact Statement

This paper presents work whose goal is to advance the field of Machine Learning. There are many potential societal consequences of our work, none which we feel must be specifically highlighted here.

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

## A. Training and Inference Details for NOMAD

### A.1. Training and Inference Algorithm

We provide the detailed algorithm pseudocode for both the training and inference procedures of the NOMAD framework in Algorithm 1 and Algorithm 2.

### A.2. Validation for the Disentangled Training Scheme

As shown in Table 5, the disentangled training scheme outperforms the entangled variant by 4 points on interPlan and 3 points on Test14-Hard(R). Furthermore, Figure 5 presents smoother convergence for the disentangled training scheme (NOMAD, blue curve), while entangled training (red dash) curve exhibits oscillations consistent and higher convergence loss.

*Table 5.* **Empirical validation for the disentangled training scheme**. The entangled variant trains its scene encoder and diffusion planner with an added loss, while DPMM is still inferred by memoVB.

|  | interPlan | Val14 (R) | Val14 (NR) | Test14-Random (R) | Test14-Random (NR) | Test14-Hard (R) | Test14-Hard (NR) |
|---|---|---|---|---|---|---|---|
| NOMAD | 70 | 95 | 94 | 93 | 88 | 85 | 78 |
| Entangled | 66 | 93 | 91 | 92 | 86 | 82 | 76 |

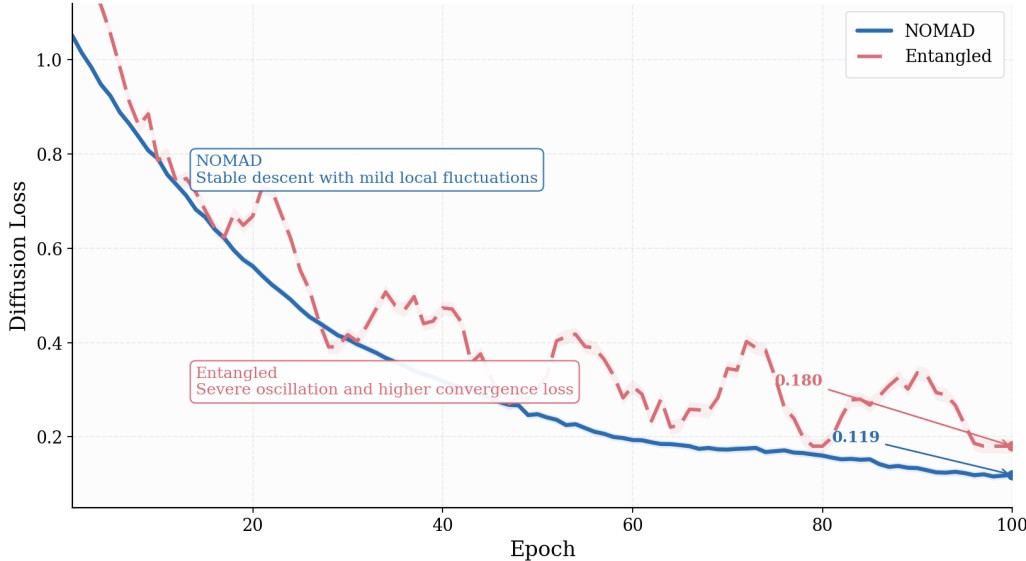

*Figure 5.* **Training loss curve comparison between disentangled and entangled scheme**. The disentangled scheme (NOMAD) achieves a stable descent characterized by mild local fluctuations, whereas the entangled variant exhibits pronounced oscillation and converges to a higher loss.

## B. Variational Inference for Dirichlet Process Mixture Model

To estimate the posterior distribution of the Dirichlet Process Mixture Model (DPMM), we employ a variational inference (VI) framework. VI reformulates the inference problem as an optimization task by introducing a tractable variational distribution $q(\mathbf{v}, \boldsymbol{\theta}, \boldsymbol{\beta})$, which approximates the true, intractable posterior $p(\mathbf{v}, \boldsymbol{\theta}, \boldsymbol{\beta}|\mathbf{z})$.

We adopt the mean-field assumption, where the latent variables, including cluster assignments $v_n$, atom parameters $\theta_k$, and stick-breaking proportions $\beta_k$, are considered mutually independent. The variational distribution is factorized as:

$$q(\mathbf{v}, \boldsymbol{\theta}, \boldsymbol{\beta}) = \prod_{n=1}^{N} q(v_n|\hat{r}_n) \prod_{k=1}^{K} q(\beta_k|\hat{a}_k, \hat{b}_k) q(\theta_k|\hat{\lambda}_k) \tag{8}$$

---

**Algorithm 1** NOMAD Training

---

**Require:** Scenario stream $\mathcal{S} = \{\mathcal{D}_1, \mathcal{D}_2, \ldots\}$; scene encoder $f_\psi$; DPMM with initial $K$; diffusion planner $\epsilon_\phi$; alternation interval $I$; concentration parameter $\alpha$

**Ensure:** Trained parameters $\psi$, DPMM state $\{\hat{r}, \hat{a}, \hat{b}, \hat{\lambda}\}$, planner parameters $\phi$

1: Initialize scene encoder $f_\psi$, diffusion planner $\epsilon_\phi$
2: Initialize DPMM: $\hat{a}_k, \hat{b}_k, \hat{\lambda}_k$ for $k = 1, \ldots, K$

3: **for** each new scenario $\mathcal{D}_t$ in stream $\mathcal{S}$ **do**
4:    **// — Phase 1: Upper Part (Disentangled) —**
5:    **for** epoch $= 1, \ldots, E_{\text{upper}}$ **do**
6:      **// Step A: Update DPMM (encoder frozen)**
7:      Compute embeddings: $z_i = f_\psi(O_i), \forall(O_i, \tau_i) \in \mathcal{D}_t$
8:      Update local params via memoVB: $\hat{r}_{ik} \propto \exp(\mathbb{E}_q[\log \pi_k] + \mathbb{E}_q[\log F(z_i|\theta_k)])$
9:      Accumulate sufficient statistics per batch: $\hat{N}_k = \sum_i \hat{r}_{ik}, \quad s_k(z) = \sum_i \hat{r}_{ik} T(z_i)$
10:     Update global params: $\hat{a}_k, \hat{b}_k, \hat{\lambda}_k$ from $\hat{N}_k, s_k$
11:     **// Birth move (every $I_b$ epochs)**
12:     **if** poorly fit subsamples $\mathbf{x}'$ exist **then**
13:       Fit proposal DPMM on $\mathbf{x}'$ with $K'$ clusters
14:       **if** $\text{ELBO}(K + K') > \text{ELBO}(K)$ **then**
15:         Accept: $K \leftarrow K + K'$
16:       **end if**
17:     **end if**
18:     **// Merge move (every $I_m$ epochs)**
19:     **for** candidate pair $(k_a, k_b)$ **do**
20:       **if** $\text{ELBO}(K - 1) > \text{ELBO}(K)$ **then**
21:         Merge $k_a, k_b$: $K \leftarrow K - 1$
22:       **end if**
23:     **end for**

24:     **// Step B: Update Encoder (DPMM frozen)**
25:     **for** each mini-batch $\{(O_i, \tau_i)\}$ from $\mathcal{D}_t$ **do**
26:       Compute $z_i = f_\psi(O_i)$
27:       Obtain cluster assignment $v_i = \arg\max_k \hat{r}_{ik}$
28:       Retrieve cluster params $\mu_{v_i}, \Sigma_{v_i}$ from DPMM
29:       Generate $z_{i,v_i}$ via reparameterization trick: $z_{i,v_i} = \mu_{v_i} + \Sigma_{v_i}^{1/2} \cdot \xi, \quad \xi \sim \mathcal{N}(0, I)$
30:       Compute $\mathcal{L}_{\text{KL}}$ via KL divergence and update $\psi$ via backpropagation
31:     **end for**
32:   **end for**

33:   **// — Phase 2: Lower Part (Disentangled) —**
34:   **// Generative Replay**
35:   $\phi_{\text{old}} \leftarrow \phi, \mathcal{D}_{\text{replay}} \leftarrow \varnothing$
36:   **for** each previously learned cluster $k = 1, \ldots, K_{\text{old}}$ **do**
37:     Sample $z_{\text{rep}} \sim \mathcal{N}(\mu_k, \Sigma_k)$ from cluster $k$, set $c_{\text{rep}} = k$, generate $\tau_{\text{rep}}$ using frozen $\epsilon_{\phi_{\text{old}}}$ conditioned on $(z_{\text{rep}}, c_{\text{rep}})$
38:     $\mathcal{D}_{\text{replay}} \leftarrow \mathcal{D}_{\text{replay}} \cup \{(z_{\text{rep}}, c_{\text{rep}}, \tau_{\text{rep}})\}$
39:   **end for**
40:   **// Train on combined dataset**
41:   **for** each mini-batch from $\mathcal{D}_t \cup \mathcal{D}_{\text{replay}}$ **do**
42:     Compute $z = f_\psi(O)$, and $c = \arg\max_k \hat{r}_k(z)$ from DPMM
43:     Sample $n \sim \text{Uniform}\{1, \ldots, N\}, \epsilon \sim \mathcal{N}(0, I)$, and compute noisy trajectory $\tau_n$ via forward diffusion
44:     Embed $c$ and add to timestep: $e_c = \text{Embed}(c); e_n = \text{TimestepEnc}(n), \text{cond} = e_c + e_n$    *// AdaLN conditioning*
45:     Compute loss $\mathcal{L}_{\text{diff}}$ and update $\phi$ via backpropagation
46:   **end for**
47: **end for**

---

---

**Algorithm 2** NOMAD Inference

---

**Require:** Observation $O_t$; trained encoder $f_\psi$; DPMM state; trained planner $\epsilon_\phi$; guidance scale $\gamma$; energy function $\mathcal{J}(\cdot)$; diffusion steps $N$
**Ensure:** Safe trajectory $\tau_0$

1: **// — Step 1: Scene Encoding —**
2: Compute scene embedding: $z = f_\psi(O_t)$

3: **// — Step 2: DPMM Cluster Assignment —**
4: Compute responsibilities for each cluster $k$: $\hat{r}_k \propto \exp\big(\mathbb{E}_q[\log \pi_k] + \mathbb{E}_q[\log F(z \,|\, \theta_k)]\big)$
5: Assign cluster: $c = \arg\max_k \hat{r}_k$

6: **// — Step 3: Conditional Diffusion Generation —**
7: Sample initial noise: $\tau_N \sim \mathcal{N}(0, I)$
8: Compute conditioning:
       $e_c = \text{Embed}(c); \quad \text{cond}(n) = e_c + \text{TimestepEnc}(n)$

9: **for** $n = N, N-1, \ldots, 1$ **do**
10:     **// Predict denoised mean**
11:     $\mu_\phi = \mu_\phi(\tau_n, \text{cond}(n), z)$
12:     $\Sigma_\phi = \Sigma_\phi(\tau_n, n)$

13:     **// — Step 4: Classifier Guidance —**
14:     Compute safety gradient: $g = \nabla_{\tau_n} \mathcal{J}(\tau_n)$
15:     Adjust mean: $\hat{\mu} = \mu_\phi - \gamma \cdot \Sigma_\phi \cdot g$

16:     **// Reverse diffusion step**
17:     **if** $n > 1$ **then**
18:         $\tau_{n-1} = \hat{\mu} + \Sigma_\phi^{1/2} \cdot \xi, \quad \xi \sim \mathcal{N}(0, I)$
19:     **else**
20:         $\tau_0 = \hat{\mu}$
21:     **end if**
22: **end for**

23: **return** Trajectory $\tau_0$

---

where $K$ is a truncated number of components chosen to be sufficiently large. To optimize the parameters $\{\hat{r}, \hat{a}, \hat{b}, \hat{\lambda}\}$, we maximize the Evidence Lower Bound (ELBO):

$$\text{ELBO}(q) = \mathbb{E}_q[\log p(\mathbf{z}, \mathbf{v}, \boldsymbol{\theta}, \boldsymbol{\beta})] - \text{KL}(q(\mathbf{v}, \boldsymbol{\theta}, \boldsymbol{\beta}) \| p(\mathbf{v}, \boldsymbol{\theta}, \boldsymbol{\beta})) \tag{9}$$

The ELBO is maximized using a coordinate ascent algorithm, iteratively updating local parameters $(\hat{r}_{nk})$ and global parameters $(\hat{a}_k, \hat{b}_k, \hat{\lambda}_k)$. For scalability with large datasets, we utilize memoized Variational Bayes (memoVB) (Hughes & Sudderth, 2013). This approach computes summary statistics $\hat{N}_k$ and $s_k(z)$ via batch-wise processing, defined as:

$$\text{ELBO}(q) \approx \sum_{k=1}^{K} \left[ \mathbb{E}_q[\theta_k]^\top s_k(z) - \hat{N}_k[a(\theta_k)] \right] + \text{Prior Terms} \tag{10}$$

The memoVB framework features batch processing capabilities. It leverages the additive property of sufficient statistics, computing local statistics based solely on the data from the current batch during each iteration, and subsequently aggregating them into the global statistics. Besides, to ensure computability, standard VB typically forces a truncation of $K$, causing the DPMM to lose its characteristics of a non-parametric model. The memoVB inference includes both birth and merge moves:

- **Birth Move**: As the model processes the current batch, if it identifies certain data points $x'$ that fit poorly within the existing $K$ clusters, it utilizes these outliers to separately fit a small-scale DPMM (hypothesizing the existence of $K'$ new clusters). If the algorithm determines that incorporating these $K'$ new clusters into the main model results in

an increase in the overall ELBO, they are formally accepted. At this point, the total number of clusters dynamically updates to $K + K'$.

- **Merge Move**: Conversely, the algorithm also periodically evaluates the existing clusters. If it determines that merging two specific clusters would improve the overall ELBO, it fuses them, thereby reducing the number of clusters to $K - 1$. It consolidates redundant clusters when ELBO improves, naturally bounding cluster growth.

## C. Design of the Mixture of Diffusion Experts Planner

As illustrated in Figure 1, the detailed design of the Mixture of Diffusion Experts Planner is presented.

**Conditioning Initialization**: The planner operates as a conditional diffusion model that iteratively refines the trajectory. The input at diffusion step $n$ is denoted as $\tau^{(n)}$, which concatenates the historical horizon $T_h$ and the noisy future horizon $T_f$. To guide the denoising process, the model integrates two key conditioning signals: the diffusion timestep $n$ and the specific expert identity determined by the cluster index $c$. The timestep $n$ is processed by a Timestep Encoder, while the cluster index $c$ is mapped through a linear embedding layer. These two representations are summed element-wise to form a unified conditioning vector. This vector is responsible for modulating the feature channels via Scale Shift operations throughout the network, effectively functioning as an Adaptive Layer Normalization (AdaLN) mechanism (Peebles & Xie, 2023).

**Dual-Stage Attention Mechanism**: The core backbone consists of $L$ stacked layers, each containing a self-attention block and a cross-attention block to capture both intra-trajectory dependencies and inter-modal context.

1. **Self-Attention Block**: The trajectory features first pass through a Layer Normalization (LN) and a Scale Shift module before entering the Multi-Head Attention (MHA) mechanism. This allows the model to capture temporal dependencies within the trajectory sequence. The output is stabilized using a Scale operation, followed by a second sequence of LN, Scale Shift, Feed-Forward Network (FFN), and a final Scale operation.

2. **Cross-Attention Block**: To incorporate environmental constraints, the refined features interact with the scene context. The scene embedding $z_{\text{scene}}$, derived from the Hierarchical Scene Encoder, serves as the Key ($K$) and Value ($V$) pairs. The trajectory features act as the Query ($Q$) in the Multi-Head Cross-Attention (MHCA) module. This interaction allows the planner to attend to relevant map and agent features encoded in $z_{\text{scene}}$.

**Trajectory Denoising and Reconstruction**: After passing through $L$ layers of iterative refinement, the feature representation undergoes a final processing stage. This stage comprises a sequence of LN, Scale Shift, and FFN to align the features for the output space. Finally, a dedicated Denoising head projects the latent features back to the trajectory dimension, predicting the denoised trajectory $\tau^{(n-1)}$ for the previous timestep. This recursive process gradually eliminates noise to generate a physically feasible and context-aware path.

## D. Details of Classifier Guidance for Safety and Comfort

Following (Zheng et al., 2025), we provide the mathematical formulations of the different energy functions.

- **Target Speed Maintenance**. The energy function is defined as a measure of the discrepancy between the average speed of the generated trajectory and the specified target speed range:

$$\mathcal{J}_{\text{TSM}} = \max\left(\frac{\mathrm{d}\tau_n}{\mathrm{d}n} - v_{\text{low}},\, 0\right)^2 + \max\left(v_{\text{high}} - \frac{\mathrm{d}\tau_n}{\mathrm{d}n},\, 0\right)^2, \tag{11}$$

where $v_{\text{low}}$ and $v_{\text{high}}$ are the lower and higher bound of speed.

- **Comfort**. We compute the deviation of each point from the comfort threshold as an energy function for longitudinal jerk, disregarding instances where comfort requirements are already satisfied.

$$\mathcal{J}_{\text{CF}} = \mathbb{E}\left[\max\left(\left(j_{\max} - \left|\frac{\mathrm{d}^3\tau_n}{\mathrm{d}n^3}\right|\right)\Delta n^3,\, 0\right)^2\right] \tag{12}$$

Where $j_{\max}$ is the maximum longitude jerk limit.

- **Staying within Drivable Area**. A differentiable cost map M is constructed ([Cheng et al., 2024a](#)), enabling per-timestamp computation of the signed distance representing lane boundary violations by the ego vehicle. The energy function is defined as:

$$\mathcal{J}_{\text{SwDA}} = \frac{1}{\omega_d} \cdot \frac{\sum_n \Psi(\omega_d \cdot \mathbf{M}(\tau_n))}{\sum_n \mathbb{I}_{\mathbf{M}(\tau_n)>0} + \varepsilon} \tag{13}$$

where $\Psi(x) = e^x - x$.

## E. Continual Learning Metrics

Continual learning aims to maintain strong performance across all tasks. Accordingly, the average closed-loop score (ACLS) is adopted as the primary evaluation metric, as it directly captures this objective and serves as the central target of continual learning optimization.

$$\text{ACLS}_t = \frac{1}{t} \sum_{l=1}^{t} s_{t,l} \tag{14}$$

Catastrophic forgetting is a central challenge in continual learning, manifesting as performance degradation on previously learned tasks. This effect is quantified by the backward transfer (BWT) metric, which measures the aggregated performance difference between the current score and the score on earlier tasks across all task pairs. BWT thus captures model stability, with smaller performance drops indicating greater resistance to disruption by newly learned tasks.

$$\text{BWT}_t = \frac{1}{t-1} \sum_{l=1}^{t-1} (s_{t,l} - s_{l,l}) \tag{15}$$

Continual learning algorithms also affect performance on the current task, as constraints imposed by previously learned tasks influence training. A model trained solely on the current task is defined as the reference model and represents the upper-bound performance achievable without such constraints. The forward transfer (FWT) metric quantifies this effect by measuring the performance difference between the continual learning model and the reference model. FWT thus reflects model plasticity, with smaller gaps indicating a stronger ability to learn new tasks efficiently.

$$\text{FWT}_t = \frac{1}{t-1} \sum_{l=2}^{t} (s_{l,l} - s_l^I) \tag{16}$$

## F. Description of Planning Baselines

We summarize the representative baseline methods for closed-loop trajectory planning as follows:

1. **IDM** ([Treiber et al., 2000](#)): a car following model for safe and realistic traffic flow simulation that focuses on accident prevention and maintains a safe distance from the leading vehicle through adaptive speed control.

2. **Urban Driver** ([Scheel et al., 2022](#)): a policy gradient method that can efficiently learn and generalize imitative driving policies for complex urban scenarios with a differentiable simulator and mid-level representations.

3. **PDM-Closed** ([Dauner et al., 2023](#)): a rule-based planner inspired by model predictive control that generates candidate trajectories through forecasting and proposal, evaluates them via simulation and scoring, and selects the trajectory with the highest score.

4. **GameFormer** ([Huang et al., 2023](#)): a learning based planner that adopts a transformer-based architecture together with hierarchical game theoretic reasoning to capture interactive behaviors among traffic participants.

5. **DTPP** ([Huang et al., 2024](#)): a differentiable joint training framework that couples ego conditioned motion prediction with learnable, context-aware cost evaluation in a tree-structured policy planner.

6. **STR2** ([Sun et al., 2025](#)): a mixture-of-experts autoregressive motion planner that combines ViT and causal transformers to generalize effectively across diverse urban driving scenarios.

7. **HybridLLMPlanner** (Hallgarten et al., 2024): a two-stage hybrid planner that integrates a large language model for behavior planning with PDM-Closed for motion planning.

8. **Diffusion-ES** (Yang et al., 2024a): a learning based planner that integrates diffusion modeling with evolutionary search and iteratively refines candidate trajectories to identify the optimal solution.

9. **PlanTF** (Cheng et al., 2024b): a planner grounded in imitation learning that prioritizes critical ego-vehicle features and leverages strategic data augmentation techniques to minimize error accumulation and bridge the imitation gap.

10. **Pluto** (Cheng et al., 2024a): an imitation learning–based planner that integrates longitudinal-lateral awareness, contrastive learning, and an auxiliary loss.

11. **Diffusion Planner** (Zheng et al., 2025): a transformer-based diffusion model that unifies prediction and planning to generate trajectories, with classifier guidance ensuring high-quality sampling without the need for rule-based heuristics.

12. **SAH-Drive** (Fan et al., 2025): a scenario-aware hybrid planner that combines rule-based and learning-based approaches via a dual-timescale decision mechanism to improve generalization and efficiency in autonomous driving trajectory planning.

13. **LiloDriver** (Yao et al., 2025): a lifelong learning motion planner that leverages large language models and memory-augmented reasoning to adaptively handle long-tail driving scenarios and mitigate catastrophic forgetting.

14. **CarPlanner** (Zhang et al., 2025a): a consistent auto-regressive reinforcement learning trajectory planner with a generation-selection framework and expert-guided reward.

15. **DAPlanner** (Zhang et al., 2025b): a dual-agent motion planning framework that leverages multi-modal large language models with trajectory generation and discrimination agents.

## G. Comparison of Continual Learning Strategy

We compare generative replay strategy against other continual learning approaches including regularization-based (elastic weight consolidation style (Kirkpatrick et al., 2017)) and architecture-based models. As illustrated in Table 6, generative replay outperforms both by 4-5 points, validating it as the best strategy for NOMAD.

*Table 6*. **Continual Learning Strategy Comparison**. Regularisation-based methods augment the loss function with an additional regularisation term, whereas architecture-based methods fix the subnetwork assigned to prior tasks and fine-tune the remaining component for the incoming tasks.

| | interPlan | Val14 (R) | Val14 (NR) | Test14-Random (R) | Test14-Random (NR) | Test14-Hard (R) | Test14-Hard (NR) |
|---|---|---|---|---|---|---|---|
| NOMAD | **70** | **95** | **94** | **93** | **88** | **85** | **78** |
| Regularization | 65 | 92 | 92 | 91 | 88 | 82 | 77 |
| Architecture | 66 | 93 | 92 | 91 | 87 | 83 | 78 |

## H. Evaluation of the Fidelity of Generative Replay Samples

As it is difficult to directly evaluate fidelity of the generative replay samples, we conduct additional experiments to provide indirect evidence:

- **NOMAD vs. w/o Replay**: Removing replay entirely drops interPlan from 70 to 66 and Test14-Hard(R) from 85 to 80.

- **NOMAD vs. Perturbing** $z$: Perturbing replay embeddings with Gaussian noise (simulating distributional drift) yields interPlan=68.

These results show that the distributional quality of replay as well as its presence is critical for preserving past knowledge. The key design insight is that generative replay operates in latent space (sampling $z$ from fitted Gaussian cluster components), so the frozen model generates trajectories conditioned on representative abstractions, avoiding raw data distribution mismatch.

*Table 7.* **Validation of the Fidelity of the Generated Trajectories**. Perturbing is performed by adding Gaussian noise $\mathcal{N}(0, 10^{-5})$ to the replay embeddings $z$.

| | interPlan | Val14 (R) | Val14 (NR) | Test14-Random (R) | Test14-Random (NR) | Test14-Hard (R) | Test14-Hard (NR) |
|---|---|---|---|---|---|---|---|
| NOMAD | **70** | **95** | **94** | **93** | **88** | **85** | **78** |
| w/o Replay | 66 | 93 | 93 | 90 | 86 | 80 | 74 |
| Perturbing $z$ | 68 | 94 | 93 | 91 | 86 | 83 | 77 |

## I. The Impact of Model Size

We analyze the impact of model size on planning performance by varying the feature dimension of NOMAD while keeping all other components unchanged. Table 8 reports the number of parameters, memory occupancy, and interPlan performance under different model configurations.

We observe that NOMAD achieves the best performance with a feature dimension of 16, reaching an interPlan score of 70 while using only 237k parameters. This corresponds to approximately 40.8% of the model parameters used by SAH-Drive, yet delivers a substantially higher long-tail performance. Reducing the feature dimension to 8 leads to a noticeable drop in performance, suggesting insufficient representational capacity for capturing diverse long-tail scenarios. Conversely, increasing the feature dimension to 32 also degrades performance despite higher capacity, indicating potential overfitting and reduced generalization in rare scenarios.

The results indicate that effective long-tail planning does not necessarily benefit from larger models. Instead, NOMAD relies on structured scene representations and memory-guided expert specialization to achieve strong performance with compact architectures. This efficiency makes NOMAD particularly suitable for real-time deployment and continual learning settings, where both computational cost and memory footprint are critical.

*Table 8.* **Impact of model parameter size**. With a feature dimension of 16, the proposed NOMAD achieves the best performance while using only 40.8% of the model parameters compared to the strongest baseline, SAH-Drive.

| Method | Feature Dimension | Model Parameters | Memory Occupancy | interPlan Score |
|---|---|---|---|---|
| | 8 | 124k | 0.48Mb | 64 |
| NOMAD (Ours) | 16 | 237k | 0.92Mb | 70 |
| | 32 | 484k | 2.07Mb | 67 |
| SAH-Drive | 32 | 581K | 2.22Mb | 64 |

## J. Visualization of Scene Embeddings and Clusters

We further visualize the temporal formation of a new lane-change cluster from 0 to 10 samples in Figure 6, showing the DPMM birth move progressively discovering and refining cluster boundaries during training. The memoVB framework includes both birth and merge moves, allowing it to decide whether a scene should form a new cluster or join an existing one based on the behavioral significance of the variation. As shown in Figure 6, although the new scene still belongs to the lane-change category, DPMM creates a new cluster for it, whereas the following scenes, though behaviorally different from the first one, are assigned to the existing Lane Change (HTD) cluster. The results confirm that the DPMM discovers meaningful scenario structure, not arbitrary latent-space partitions.

## K. Robustness and Sensitivity Analysis

We conduct a robustness analysis to evaluate the impact of partial observations on scene embeddings, wherein 10%, 20%, and 30% of input agent trajectories are randomly masked. As shown in Table 9, performance exhibits graceful degradation: under 30% trajectory masking, the interPlan score persists at 65, which still surpasses the strongest baseline, SAH-Drive (64). This finding substantiates the model's reasonable robustness to perceptual noise and partial observations.

We perform a sensitivity analysis on the interPlan benchmark by sweeping the guidance scale parameter $\gamma$ over the range $[0.5, 2.5]$. As reported in Table 10, increasing $\gamma$ enhances collision avoidance while diminishing overall driving quality (i.e.,

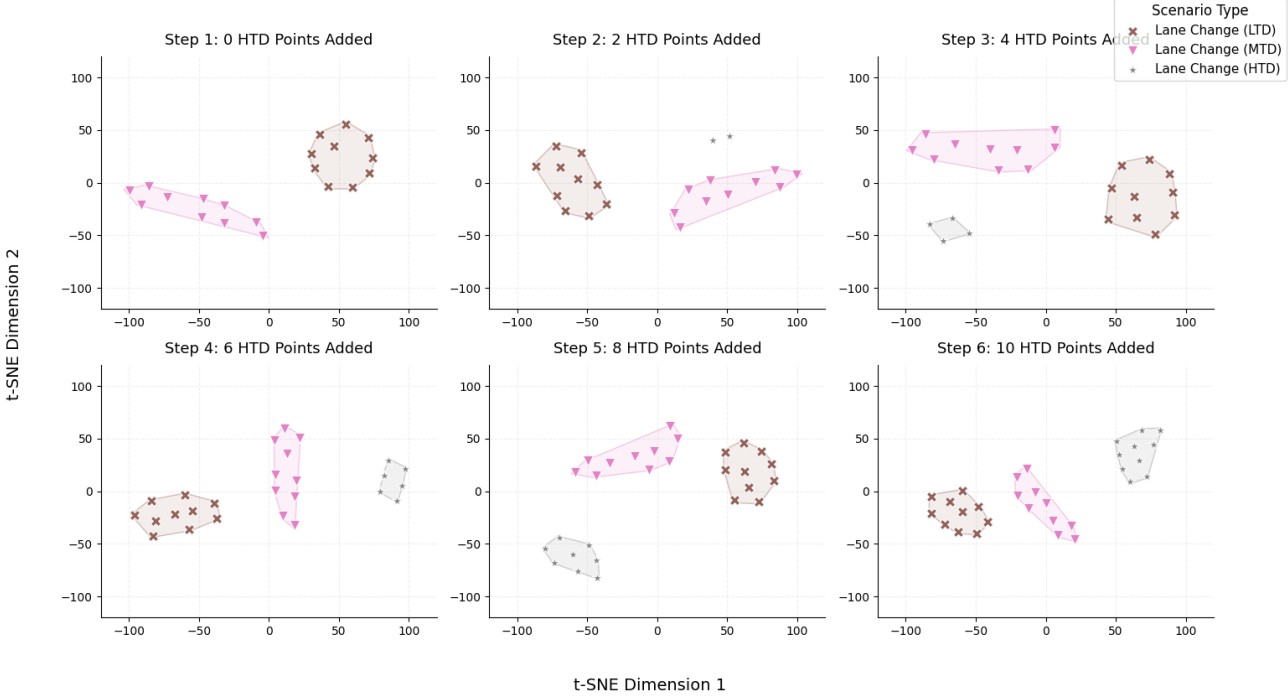

*Figure 6.* **Continuous Formation of a New Lane-Change Cluster via DPMM**. The formation of a new lane-change cluster (HTD) from 0 to 10 samples illustrates both the birth-move behavior of DPMM and the shape retention of existing clusters (LTD and MTD).

*Table 9.* **Robustness Analysis under Partial Observations on Scene Embeddings**.

| Masking | interPlan | Val14 (R) | Val14 (NR) | Test14-Random (R) | Test14-Random (NR) | Test14-Hard (R) | Test14-Hard (NR) |
|---|---|---|---|---|---|---|---|
| 0% (full) | **70** | **95** | **94** | **93** | **88** | **85** | **78** |
| 10% | 68 | 94 | 94 | 92 | 87 | 85 | 77 |
| 20% | 67 | 94 | 93 | 92 | 87 | 84 | 76 |
| 30% | 65 | 93 | 91 | 90 | 86 | 84 | 76 |

yields more conservative trajectories). The optimal trade-off occurs at $\gamma = 1.0$, consistent with prior work (Zheng et al., 2025). The observed trend is both monotonic and predictable, implying that practical tuning is straightforward.

*Table 10.* **Sensitivity Analysis of the Guidance Scale Parameter**.

| $\gamma$ | 0.5 | 1.0 | 1.5 | 2.0 | 2.5 |
|---|---|---|---|---|---|
| Collision avoidance | 86 | 92 | 93 | 93 | **94** |
| score | **71** | 70 | 68 | 67 | 65 |

## L. Failure Case

Figure 7 illustrates representative failure cases in parking lot scenarios. Compared to structured urban driving environments, parking lots are characterized by sparse lane semantics, weak traffic rules, and limited interactions with dynamic agents. As a result, both NOMAD and the strongest baseline, SAH-Drive, occasionally generate incorrect trajectories in such settings.

Specifically, the lack of reliable motion cues from surrounding agents and the absence of strong map priors make it difficult to infer driving intent and scene context. While NOMAD benefits from memory-guided continual learning and produces more reasonable trajectories than SAH-Drive, the underlying scene embeddings remain ambiguous, leading to inaccurate memory assignment and suboptimal trajectory generation. This failure mode highlights a fundamental limitation of memory-guided

planning when semantic structure and dynamic context are insufficient.

Therefore, handling parking-lot-like scenarios may require richer spatial reasoning, explicit intent inference, or integration with dedicated low-speed maneuvering modules. We leave the exploration of such extensions to future work.

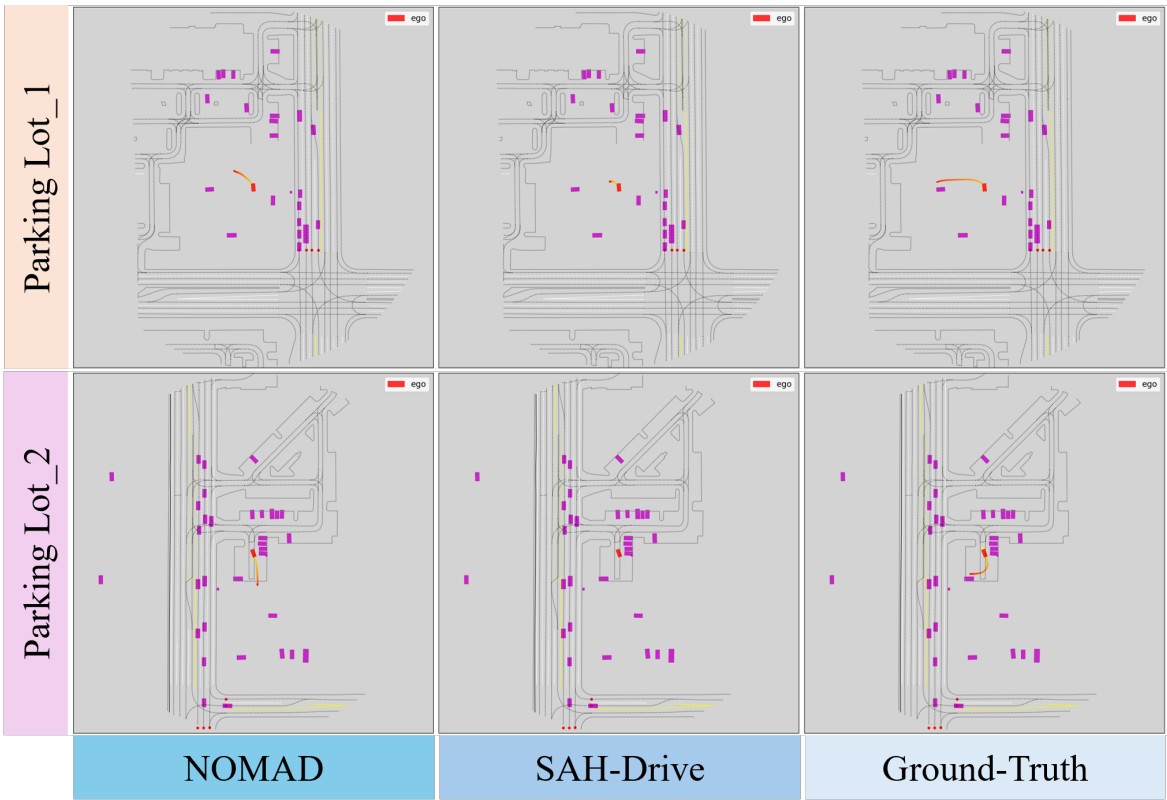

*Figure 7.* **Visualization of failure cases in the parking lot scenario**. Due to the limited cues from moving objects in this setting, both our NOMAD method and the best-performing baseline, SAH-Drive, generate incorrect trajectories. Although NOMAD outperforms SAH-Drive owing to memory-guided continual learning, this scenario remains challenging and warrants further investigation.

