# OpenReview forum: "NOMAD: Lifelong Trajectory Planning via Non-Parametric Bayesian Memory-Adaptive Diffusion Experts"
_ICML.cc/2026/Conference — ICML 2026 regular_

### Official Review · Reviewer_TwH3 · 2026-03-11

**Soundness:** 2
**Presentation:** 3
**Significance:** 3
**Originality:** 2
**Overall Recommendation:** 4
**Confidence:** 4

**Summary:**

The paper proposes NOMAD, a lifelong trajectory planning framework for open-world autonomous driving. It combines a hierarchical scene encoder, a non-parametric Bayesian memory that clusters scene embeddings into scenario contexts, and a memory-conditioned diffusion planner, with generative replay to mitigate forgetting during continual adaptation. Experiments on nuPlan/interPlan show strong long-tail closed-loop performance while maintaining competitive results on regular driving scenarios.

**Compliance With Llm Reviewing Policy:**

Affirmed.

**Key Questions For Authors:**

1. Could the authors clarify more explicitly how the non-parametric memory and the generative replay mechanism interact? As described in the paper, replay samples are generated by sampling scene embeddings from the fitted memory components and then using the frozen old model to synthesize trajectories. If replay is ultimately driven by scene-level memory, how realistic and behaviorally faithful are the generated trajectories, especially for preserving fine-grained planning knowledge?

2. The paper mainly motivates its design by highlighting the latency and retrieval dependence of LLM-based planners. It would be helpful to clarify whether the main advantage of the proposed framework is primarily computational, or whether the authors also view it as a better modeling choice for continual closed-loop planning than recent LLM-based planners with stronger reasoning and prompt-based adaptability.

**Limitations:**

Yes

**Strengths And Weaknesses:**

Pros:

1.	The paper targets an important and practically relevant problem. Handling long-tail scenarios under continual adaptation is a meaningful challenge for autonomous driving, and the paper frames this setting clearly rather than focusing only on static benchmark gains.

2.	The paper reports results on both regular and long-tail benchmarks, includes continual-learning evaluation, and provides ablations showing that memory, scene encoding, and memory-guided diffusion each contribute to performance.

3.	A technical strength of the paper is that it integrates scenario-aware memory with a conditional diffusion planner in a unified framework, and the ablation results suggest that this coupling is not merely cosmetic but contributes meaningfully to long-tail planning performance.



Cons:

1.  The proposed memory is based on clustering latent scene embeddings with a DPMM, but the paper does not sufficiently show that the resulting clusters correspond to meaningful planning scenarios rather than latent-space partitions. It is also unclear how this clustering behaves as continual learning proceeds and the number of encountered scenarios keeps growing, especially when small but behaviorally important scene changes may induce new clusters.
2.  The paper explicitly uses a single diffusion backbone conditioned on memory context, which appears closer to context-modulated diffusion than to a standard multi-expert MoE design.
3.  The paper motivates diffusion mainly by its ability to model multi-modal trajectories, but does not sufficiently discuss why this formulation is preferable to policy-based planning or other sequential decision-making approaches for continual adaptation. More broadly, the method description is detailed, while the modeling motivation for combining clustering, diffusion, and replay is less clearly justified.

---

> ### Author Rebuttal · Authors · 2026-03-30
>
> We address each concern below.
>
> **[W1]**
>
> We have added two new visualizations with anonymous links:
>
> (1) **t-SNE of interPlan scene embeddings** ([link](https://anonymous.4open.science/r/NOMAD_Rebuttal/t-SNE%20Visualization%20of%20interPlan%20Scene%20Embeddings%20Generated%20by%20DPMM.png)) shows 8 long-tail scenario types forming clearly separated clusters, with quantitative cluster purity 100% against ground-truth labels. Semantically, each cluster corresponds to a distinct driving behavior (e.g., construction site, jaywalker, overtake).
>
> (2) We visualize the **temporal formation of a new lane-change cluster** from 0 to 10 samples ([link](https://anonymous.4open.science/r/NOMAD_Rebuttal/Continuous%20Formation%20of%20a%20New%20Lane-Change%20Cluster%20via%20DPMM.png)), showing the DPMM birth move progressively discovering and refining cluster boundaries during training. These results confirm that the DPMM discovers meaningful scenario structure, not arbitrary latent-space partitions.
>
> Regarding **small but behaviorally important changes**, the memoVB framework includes both birth and merge moves (see Reply to **Reviewer CQ6d-Q1**), allowing it to decide whether a scene should form a new cluster or join an existing one based on the behavioral significance of the variation. As shown in the above Visualization (2), although the new scene still belongs to the lane-change category, DPMM creates a new cluster for it, whereas the following scenes, though behaviorally different from the first one, are assigned to the existing Lane Change (HTD) cluster.
>
> **[W2]**
>
> We consider the reviewer's characterization and will adopt the clearer term "**mixture of conditional diffusion experts**" in the revision. The analogy comes from **Gaussian Mixture Models**: just as a GMM uses shared exponential-family forms with per-component parameters, our diffusion model uses a shared denoising backbone where the cluster assignment $c$ modulates feature channels via Adaptive Layer Normalization (AdaLN). To prevent interference across disparate scenarios, the model receives dual contexts: the continuous embedding $z$ captures **instance-specific scene features**, while the discrete cluster assignment $c$ provides **scenario-level expert information**. Our ablation (Table 3, "w/ Normal Diffusion") shows that removing the cluster-conditioning degrades interPlan by 4.3%, confirming that the conditioning mechanism meaningfully specializes the denoising process.
>
> **[W3 & Q2]**
>
> We provide **new baseline comparisons** against RL-based and LLM-based planners:
>
> |                 | interPlan  | Val14 (R)  | Val14 (NR)  | Test14-Random (R)  | Test14-Random (NR)  | Test14-Hard (R)  | Test14-Hard (NR) |
> |-|-|-|-|-|-|-|-|
> | NOMAD           |         **70** |         **95** |          **94** |                 **93** |                  88 |               **85** |               **78** |
> | CarPlanner (RL, CVPR 2025) |         60 |         88 |          86 |                 91 |                  **90** |               80 |               72 |
> | DAPlanner (LLM, Applied Soft Computing 2025) |         55 |         87 |          85 |                 84 |                  85 |               81 |               70 |
>
> NOMAD offers three key advantages:
>
> (1) **Efficiency**: it runs at about 1.5 s/frame, whereas LLM planners require much heavier inference.
>
> (2) **Multi-modal expressiveness**: its diffusion experts model multi-modal trajectory distributions, which are essential in long-tail scenarios with multiple valid responses, such as stopping, swerving, or accelerating. RL policies typically favor a single mode, and LLM planners, although semantically strong, often exhibit modality bias and hallucination.
>
> (3) **Continual learning compatibility**: with DPMM and generative replay, NOMAD supports continual learning via a finite set of memory clusters, a structural advantage for mitigating catastrophic forgetting that RL and LLM planners do not naturally have.
>
> **[Q1]**
>
> For details on the training pipeline and module interactions, please refer to **Reviewer ruF3-W1**.
>
> As you noted, replay trajectories are sampled from the learned memory clusters. To validate the realism of the generated trajectories, we conduct additional experiments and observe two key findings (see the table in Reply to **Reviewer 56eG-W3(1)** for details):
> - **NOMAD v.s. w/o Replay**: Removing replay entirely drops interPlan from 70 to 66 and Test14-Hard(R) from 85 to 80.
> - **NOMAD v.s. Perturbing $z$**: Perturbing replay embeddings with Gaussian noise (simulating distributional drift) yields interPlan=68, showing that the distributional quality of replay as well as its presence is critical for preserving past knowledge.
>
> The key design insight is that generative replay operates in latent space (sampling $z$ from fitted Gaussian cluster components), so the frozen model generates trajectories conditioned on **representative abstractions**, avoiding raw data distribution mismatch.

---

> > ### Author Rebuttal · Reviewer_TwH3 · 2026-04-02
> >
> > Thanks for the rebuttal. The authors solved most of my problem. Based on the quality and novelty of the paper, I think my previous score is reasonable, and I will maintain my score.

---

> > > ### Author Response · Authors · 2026-04-02
> > >
> > > Dear Reviewer TwH3,
> > >
> > > &nbsp;
> > >
> > > We sincerely thank you for your thoughtful review, your positive feedback, and the time and effort you invested in evaluating our paper. We truly appreciate your engagement and support.
> > >
> > > If you feel that your concerns have been fully resolved, we would be sincerely grateful if you would consider improving your overall score or soundness/originality.
> > >
> > > &nbsp;
> > >
> > > Sincerely,
> > >
> > > Authors of Paper 3895

---

### Official Review · Reviewer_CQ6d · 2026-03-12

**Soundness:** 3
**Presentation:** 4
**Significance:** 3
**Originality:** 3
**Overall Recommendation:** 4
**Confidence:** 4

**Summary:**

The authors propose a novel architecture, NOMAD, that combines non-parametric Bayesian memory with diffusion-based trajectory generation. A lifelong trajectory-planning framework is proposed to enable autonomous vehicles to continually adapt to rare long-tail scenarios in open-world environments while avoiding catastrophic forgetting. Specifically, the framework maps continuous driving scenes into a dynamically expanding discrete memory using a Dirichlet Process Mixture Model (DPMM), which clusters scenarios and provides contextual conditioning signals to a diffusion-based planner that effectively acts as a mixture-of-experts trajectory generator.

**Compliance With Llm Reviewing Policy:**

Affirmed.

**Key Questions For Authors:**

1. Memory Growth Control and Memory Retrieval Efficiency
As the number of clusters increases over time, how does the system maintain efficient memory retrieval during inference?
Since the DPMM allows the memory to grow dynamically, how does the framework control the long-term growth of clusters when deployed over extended periods?

2. Scene Encoder Robustness
How sensitive is the clustering mechanism to perturbations in scene embeddings, such as perception noise or partial observations?

3. Guidance Parameter Sensitivity
How does the classifier guidance scale affect safety–performance trade-offs?

**Limitations:**

-

**Strengths And Weaknesses:**

The paper presents a technically interesting and well-executed framework for lifelong trajectory planning in autonomous driving. The combination of non-parametric memory, diffusion-based planning, and generative replay is novel and shows strong empirical performance.
However, several aspects, such as particularly the reliance on scene encoder quality, the limited robustness analysis, and the lack of real-world validation, leave open questions regarding the system’s practical deployment.

Strengths:
1. The integration of non-parametric Bayesian memory (DPMM) with a conditional diffusion-based planner seems a technically elegant design. The use of DPMM allows the system to dynamically expand its scenario memory as new driving situations emerge, while the diffusion model provides a flexible generative mechanism for multi-modal trajectory synthesis. This architecture provides a principled solution to the stability–plasticity trade-off that arises in continual learning settings.
2. The proposed generative replay mechanism offers a practical alternative to traditional replay buffers that require storing raw historical data. By synthesizing representative trajectories from previously learned clusters using a frozen model, the framework mitigates catastrophic forgetting while avoiding storage overhead and potential privacy issues. This paper provides a Practical Solution to Catastrophic Forgetting.
3. The empirical evaluation on the nuPlan benchmark shows that NOMAD consistently outperforms strong baselines across multiple scenario types.
4. The framework demonstrates favorable computational efficiency and maintains real-time inference capabilities.

Weaknesses & Limitations:
1. The proposed framework struggles in environments lacking strong semantic or structural cues, such as open parking lots. In such cases, the absence of clear map structure or motion priors can lead to unreliable memory assignment and degraded trajectory generation performance.
2. The effectiveness of the DPMM-based memory allocation relies heavily on the quality of the latent scene embeddings produced by the hierarchical scene encoder. However, the paper provides limited analysis of the system's sensitivity to variations in these embeddings.

---

> ### Author Rebuttal · Authors · 2026-03-30
>
> We sincerely thank the reviewer for the positive assessment of NOMAD's technically elegant design, practical solution to catastrophic forgetting, strong empirical performance, and favorable computational efficiency. We address the remaining questions below.
>
> **[W1]**
>
> We agree that this is a **common limitation**. As analyzed in Appendix G, environments with sparse semantic/structural cues (e.g., open parking lots) challenge both NOMAD and baseline methods, including SAH-Drive. The lack of strong map priors and dynamic agent interactions makes reliable memory assignment difficult. We view this as a fundamental challenge for memory-guided planners and leave the exploration of richer spatial reasoning or dedicated low-speed maneuvering modules to future work.
>
> **[W2 & Q2]**
>
> We have added a new **robustness analysis** evaluating the impact of **partial observations on scene embeddings**. We randomly mask 10%, 20%, and 30% of input agent trajectories:
>
> | Masking | interPlan  | Val14 (R)  | Val14 (NR)  | Test14-Random (R)  | Test14-Random (NR)  | Test14-Hard (R)  | Test14-Hard (NR) |
> |-|-|-|-|-|-|-|-|
> | 0% (full) |         **70** |         **95** |          **94** |                 **93** |                  **88** |               **85** |               **78** |
> |   10% |         68 |         94 |          94 |                 92 |                  87 |               85 |               77 |
> |   20% |         67 |         94 |          93 |                 92 |                  87 |               84 |               76 |
> |   30% |         65 |         93 |          91 |                 90 |                  86 |               84 |               76 |
>
> Performance degrades gracefully: even with 30% trajectory masking, interPlan remains at 65, still exceeding the strongest baseline SAH-Drive (64). This confirms reasonable robustness to perception noise and partial observations.
>
> **[Q1]**
>
> Two key clarifications:
>
> (1) Inference: At inference, the scene encoder computes $z$ once, and the DPMM assigns cluster assignment $c$ via a forward pass through the variational posterior, which is a lightweight $O(K)$ operation where $K$ is typically 8 in our experiments. **No retrieval search** over stored exemplars is needed.
>
> (2) Growth control: We adopt **memoVB** (Hughes & Sudderth, 2013)) instead of standard VB here to resolve the contradiction between streaming data and infinite growth. First, memoVB features **batch processing** capabilities. It leverages the additive property of sufficient statistics, computing local statistics based solely on the data from the current batch during each iteration, and subsequently aggregating them into the global statistics. Second, to ensure computability, standard VB typically forces a truncation of $K$, causing the DPMM to lose its characteristics of a non-parametric model. The memoVB inference includes both birth and merge moves:
> - **Birth Move**: As the model processes the current batch, if it identifies certain data points $x'$ that fit poorly within the existing $K$ clusters, it utilizes these outliers to separately fit a small-scale DPMM (hypothesizing the existence of $K'$ new clusters). If the algorithm determines that incorporating these $K'$ new clusters into the main model results in an increase in the overall ELBO, they are formally accepted. At this point, the total number of clusters dynamically updates to $K + K'$.
> - **Merge Move**: Conversely, the algorithm also periodically evaluates the existing clusters. If it determines that merging two specific clusters would improve the overall ELBO, it fuses them, thereby reducing the number of clusters to $K - 1$. It consolidates redundant clusters when ELBO improves, naturally bounding cluster growth. In our experiments, the DPMM stabilized at approximately 8 active clusters. At feature dimension $d=16$, NOMAD stores only **0.92MB** total (Appendix Table 5), making memory overhead negligible.
>
> **[Q3]**
>
> We provide a new **sensitivity analysis** on the interPlan benchmark with **guidance scale parameter** $\gamma$ varying from 0.5 to 2.5:
>
> | $\gamma$                    | 0.5 | 1.0 | 1.5 | 2.0 | 2.5 |
> |-|-:|-:|-:|-:|-:|
> | Collision avoidance |  86 |  **92** |  93 |  93 |  94 |
> | score               |  **71** |  70 |  68 |  67 |  65 |
>
> Higher $\gamma$ improves collision avoidance but reduces overall driving quality (more conservative trajectories). **$\gamma=1.0$ provides the best trade-off**, consistent with prior work (Diffusion Planner, ICLR 2025). The trend is monotonic and predictable, suggesting easy tuning in practice.
>
> **References**
>
> Hughes, M.C. and Sudderth, E., 2013. Memoized online variational inference for Dirichlet process mixture models. Advances in Neural Information Processing Systems, 26.

---

> > ### Author Rebuttal · Reviewer_CQ6d · 2026-04-05
> >
> > The authors’ rebuttal has resolved a substantial part of the original technical concerns and has updated the overall assessment in a positive direction. The authors responded carefully and credibly to the technical questions, which improved confidence in the work. The contribution is considered clear and meaningful in the simulation setting. Despite the remaining limitation of the lack of real-world experiments, the reviewer’s final stance is supportive overall, although a score of approximately over 4 (4.4) still appears to be the most appropriate rating.

---

### Official Review · Reviewer_56eG · 2026-03-12

**Soundness:** 3
**Presentation:** 2
**Significance:** 3
**Originality:** 3
**Overall Recommendation:** 4
**Confidence:** 4

**Summary:**

This paper introduces NOMAD, a lifelong learning trajectory planning framework for autonomous driving that integrates non-parametric Bayesian memory with diffusion-based trajectory generation. The key idea is to map continuous driving scenes to a dynamically expanding discrete memory structure using a Dirichlet Process Mixture Model (DPMM). A conditional diffusion model serves as the continuous planner, conditioned on the retrieved memory context to specialize generation for different scenario types. To mitigate catastrophic forgetting, the framework employs a generative replay mechanism and safety constraints are enforced during inference via classifier guidance. Experiments on the nuPlan benchmark demonstrate state-of-the-art performance on long-tail scenarios while maintaining strong performance on regular driving.

**Compliance With Llm Reviewing Policy:**

Affirmed.

**Final Justification:**

I appreciate the comments during the rebuttal phase and I will maintain my recommendation.

**Key Questions For Authors:**

1. Does each diffusion expert share parameters, or are there separate parameters per cluster? If parameters are shared, how does the model avoid interference when scenarios are very different? Please clarify the Mixture-of-Experts interpretation.

2. Are the clusters semantically meaningful and consistent across training runs? More validations are encouraged.

3. Is the Hierarchical Scene Encoder updated during lifelong learning, or is it frozen after initial training? If it is frozen, how does it generalize to novel scenarios not seen during pretraining?

4. How to evaluate the fidelity of synthetic trajectories generated by the frozen model for replay? Could low-quality replay data introduce bias or degrade performance? Have does generative replay compare against other forgetting mitigation strategies?

**Limitations:**

The authors discuss their limitations in Conclusion briefly.

**Strengths And Weaknesses:**

**Strengths**

1. The use of a Dirichlet Process Mixture Model for automatic cluster discovery is elegant. It avoids heuristic clustering and predefined cluster counts, allowing the memory to expand naturally as new scenario types emerge.

2. Conditioning a diffusion planner on retrieved memory context is a clever way to achieve scenario-specialized generation without training separate models. The mixture-of-experts interpretation is well-articulated.

3. NOMAD achieves SOTA on the challenging interPlan long-tail benchmark.

**Weaknesses**

1. Mixture-of-Experts interpretation may be overstated. While the paper frames the diffusion model as a "mixture of diffusion experts," the architecture appears to be a single diffusion model conditioned on a cluster index c. This is more accurately described as a conditional diffusion model rather than a true mixture of experts. The distinction matters because a conditional model may struggle to fully specialize to disparate scenarios without parameter interference.

2. The paper assumes that the DPMM discovers meaningful scenario clusters but provides no analysis of what these clusters represent. Without this validation, it is unclear whether the memory is truly capturing scenario structure or merely overfitting to low-level perceptual features.

3. The success of generative replay hinges on the quality of synthetic trajectories from the frozen model. The paper does not evaluate the quality of replayed data or its impact on forgetting mitigation.

---

> ### Author Rebuttal · Authors · 2026-03-30
>
> We address each concern below.
>
> **[W1 & Q1]**
>
> We agree that the architecture uses a single shared backbone, in part for efficiency, and we will adopt the clearer term "**mixture of conditional diffusion experts**" in the revision. The analogy comes from **Gaussian Mixture Models**: just as a GMM uses shared exponential-family forms with per-component parameters, our diffusion model uses a shared denoising backbone where the cluster assignment $c$ modulates feature channels via Adaptive Layer Normalization.
>
> To prevent interference across disparate scenarios, the model receives dual conditioning: the continuous embedding $z$ captures **instance-specific scene features**, while the discrete cluster assignment $c$ provides **scenario-level expert information**. Our ablation (Table 3, "w/ Normal Diffusion") shows that removing the cluster-conditioning degrades interPlan by 4.3%, confirming that the conditioning mechanism meaningfully specializes the denoising process.
>
> **[W2 & Q2]**
>
> We have added two new visualizations with an anonymous link:
>
> (1) **t-SNE of interPlan embeddings** ([link](https://anonymous.4open.science/r/NOMAD_Rebuttal/t-SNE%20Visualization%20of%20interPlan%20Scene%20Embeddings%20Generated%20by%20DPMM.png)): 8 distinct long-tail scenario types form clearly separated clusters, confirming semantic alignment with ground-truth scenario categories. Quantitatively, cluster purity achieve 100% against the 8 interPlan labels. Semantically, we observe that, e.g., Cluster 1 (blue circles) captures construction site detours, Cluster 4 (red diamonds) captures jaywalker avoidance, and Cluster 7 (pink inverted triangles) captures high-density lane changes.
>
> (2) **Cluster formation dynamics** ([link](https://anonymous.4open.science/r/NOMAD_Rebuttal/Continuous%20Formation%20of%20a%20New%20Lane-Change%20Cluster%20via%20DPMM.png)): We visualize a new lane-change cluster emerging from 0 to 10 samples, showing the DPMM birth move progressively refining cluster boundaries. These results confirm that clusters are semantically meaningful and stable during continual learning.
>
> **[Q3]**
>
> The scene encoder is updated during lifelong learning via an **alternating optimization scheme** (see paper Sec. 3.4 and Appendix A). When new scenarios arrive, the encoder is fine-tuned by minimizing the KL divergence between its embedding distribution and the updated DPMM cluster components. This allows the encoder to adapt its representations to novel scenarios. The DPMM module is also updated with birth/merge moves to accommodate new patterns. Also see the overall training pipeline presented in **[Reviewer ruF3-W1]**.
>
> **[W3 & Q4]**
>
> It is difficult to directly evaluate fidelity, we add 2 new experiments to provide indirect evidence.
>
> (1) Generative replay fidelity:
> - **NOMAD vs. w/o Replay**: Removing replay entirely drops interPlan from 70 to 66 and Test14-Hard(R) from 85 to 80.
> - **NOMAD vs. Perturbing $z$**: Perturbing replay embeddings with Gaussian noise (simulating distributional drift) yields interPlan=68, showing that the distributional quality of replay as well as its presence is critical for preserving past knowledge.
>
> The key design insight is that generative replay operates in latent space (sampling $z$ from fitted Gaussian cluster components), so the frozen model generates trajectories conditioned on **representative abstractions**, avoiding raw data distribution mismatch.
>
> |                             | interPlan  | Val14 (R)  | Val14 (NR)  | Test14-Random (R)  | Test14-Random (NR)  | Test14-Hard (R)  | Test14-Hard (NR) |
> |-|-|-|-|-|-|-|-|
> | NOMAD                       |         **70** |         **95** |          **94** |                 **93** |                  **88** |               **85** |               **78** |
> | w/o Replay                  |         66 |         93 |          93 |                 90 |                  86 |               80 |               74 |
> | $z$+Gaussian noise (0, $10^{-5}$) |         68 |         94 |          93 |                 91 |                  86 |               83 |               77 |
>
> (2) Continual Learning strategy comparison: We compare generative replay against **regularization-based** (EWC-style, interPlan=65) and **architecture-based** approaches (interPlan=66). Generative replay outperforms both by 4-5 points, validating it as the best strategy for this model.
>
> |                | interPlan  | Val14 (R)  | Val14 (NR)  | Test14-Random (R)  | Test14-Random (NR)  | Test14-Hard (R)  | Test14-Hard (NR) |
> |-|-|-|-|-|-|-|-|
> | NOMAD          |         **70** |         **95** |          **94** |                 **93** |                  **88** |               **85** |               **78** |
> | Regularization |         65 |         92 |          92 |                 91 |                  88 |               82 |               77 |
> | Architecture   |         66 |         93 |          92 |                 91 |                  87 |               83 |               78 |

---

> > ### Author Rebuttal · Reviewer_56eG · 2026-04-01
> >
> > My concerns are addressed and I will maintain my initial score.

---

> > > ### Author Response · Authors · 2026-04-02
> > >
> > > Dear Reviewer 56eG,
> > >
> > > &nbsp;
> > >
> > > We sincerely thank you for your thoughtful review, your positive feedback, and the time and effort you invested in evaluating our paper. We truly appreciate your engagement and support.
> > >
> > > If you feel that your concerns have been fully resolved, we would be sincerely grateful if you would consider improving your score.
> > >
> > > &nbsp;
> > >
> > > Sincerely,
> > >
> > > Authors of Paper 3895

---

### Official Review · Reviewer_ruF3 · 2026-03-16

**Soundness:** 3
**Presentation:** 2
**Significance:** 3
**Originality:** 3
**Overall Recommendation:** 4
**Confidence:** 3

**Summary:**

The paper presents NOMAD, a method for planning in the autonomous driving space. The authors present a method that makes use of a transformer encoder producing a latent z, as well as a non-parrametric bayesian memory trained with variational inference, combined with a diffusion model with MoE, and classifier guidance. The authors also consider a technique for storing learned representations with a pretrained model, as well as classifier guidance for safety. The authros evaluate their models on the nuplan benchmark, focusing in particular on long tail events and compare their work against various other methods.

**Compliance With Llm Reviewing Policy:**

Affirmed.

**Final Justification:**

The authors report good experiments, my initial low score was based mostly on details missing. The pdf the authors provided, along with the two rebuttal comments clarified a lot on that front.

**Key Questions For Authors:**

See Strengths / Weaknesses

**Limitations:**

yes

**Strengths And Weaknesses:**

Strengths
- The authors post excellent results on the nuplan benchmark
- The combination of techniques is novel, is an interesting exploration into the use of non-parametric methods and memory into autonomous driving

Beyond these two strengths, I do have some significant concerns about this work in particular with regards to clarity. While the method is described in broad strokes, some very critical details are lacking, and while the text does contain some formulas, these are mostly general to the previous methods the authors combined. In such a complex combination of previous methods, it is imperative to have a detailed description of each of the methods, and how they're linked togehter in terms of training and inference. Some things in particular:
- The scene encoder is trained to produce z, but it is not clear whether these are trained separately, or jointly, it is also not clear what kind of a loss is used, whether the encodings are trained specifically to work together with the dirichlet process, etc.
- While the dirichlet process is an interesting addition, it is not explained beyond some general VI statements, even in the appendix it is not made clear which parts are parameterized and by what, it is also not made clear whether these are trained jointly or separately. Moreover, a dirichlet process has an infinite support, which clearly cannot be described in a classical computer, so the authors must truncate the process, which is not described either. It is also not described whether and how this is sampled from during training, and how the authors deal with passing gradients through samples from a categorical distribution.
- The authors have a fairly generic overview of diffusion modelling, but do not describe the mixture part. The authors say that conditioning on c creates a mixture of experts, but I fail to see how that happens. Do the authors run the diffusion model for all memories c? Please clarify
- It is not clear in what part of the pipeline the "generative replay for lifelong learning" fits, from the text it seems to be after training, but it's not clear whether that means a round of distillation, or whether it can be used online for inference. It is also not clear to me how this method reduces memory, since each datapoint still creates one z.
- As a bare minimum, the authors should aim to include an algorithm in appendix or better, the main text so it is easier to review the method

I hope the authors can clarify some of these methodological questions because the results the authors pose do look very good, but in it's current form I don't think the paper meets the bar for reproducibility

---

> ### Author Rebuttal · Authors · 2026-03-30
>
> We appreciate the detailed questions and address each below.
>
> **[W1]**
>
> We clarify the full training pipeline. NOMAD is separated into two disentangled parts for parameter updates:
>
> **Upper Part (Scene Encoder + DPMM)**: We adopt an alternating optimization scheme. First, the DPMM module is updated by memoVB using scene embeddings $z$ computed from the encoder. This update occurs after a fixed number of encoder training steps. Then, with the DPMM frozen, the scene encoder is updated by minimizing the KL divergence between the encoder distribution and the assigned cluster components.
>
> To compute $L_{KL}$,
> we obtain the cluster assignment $v_i = k$ of $z_i$ from the DPMM. Using the DPMM, we determine the mean $\mu_k$ and covariance $\Sigma_k$ of the assigned cluster $k$. As such, the $i$-th inference result assigned to the component $k$, $z_{ik}$, is generated through the reparametric trick.
>
> This alternating scheme eliminates the need to refit the DPMM from scratch at every step.
>
> **Lower Part (Conditional Diffusion Planner)**: The upper part parameters are frozen. We obtain $z$ and the cluster assignment $c$, then train the diffusion model by minimizing the noise prediction loss $L_{diff}$ (Eq. 3). Training data mixes current-scenario samples with generative replay samples from previously learned scenarios to mitigate catastrophic forgetting.
>
> **[W2]**
>
> We appreciate this detailed question.
>
> (1) **Parameterization of DPMM**: We used the Dirichlet Process Mixture Model to organize the latent space instead of only the Dirichlet process. The parameterization and generative process of DPMM have been provided in Section 3.4. To make it clearer:
>
> $$\pi \sim GEM(\alpha)$$
> $$\theta_k \sim \mathcal{H}(\lambda)$$
> $$v_i \sim \text{Cat}(\pi)$$
> $$z_i \sim F(\theta_{v_i})$$
>
> - $GEM(\alpha)$: Generalized Ewens distribution, equivalent to a stick-breaking process (Eq. 1) in DPMM
> - $\pi$: Infinite-dimensional weight vector that sums to 1
> - $\mathcal{H}$: Base distribution as priors
> - $\theta_k$: Distribution parameters of the $k$-th cluster
> - $v_i$: Clustering assignment index for the $i$-th data point
> - $\text{Cat}(\cdot)$: Categorical distribution, sampling from infinite possible classes
> - $F()$: observation distribution, generate the final data $z_i$
>
> (2) **Truncation**: Standard VB uses a mean-field approximation but requires traversing the full dataset per iteration, which is infeasible for streaming data. In this case, we typically impose a sufficiently large upper bound $K$ on cluster count, e.g., for $k > K$, the selection probability is 0.
>
> (3) **Dynamic adjustment via memoVB** (Hughes & Sudderth, 2013): To preserve the non-parametric nature despite truncation, we use memoized Variational Bayes (memoVB), which processes data in batches via additive sufficient statistics. Besides, the memoVB introduces birth moves (fitting outliers to $K'$, e.g., $K'=1$ as we train new scenarios one by one, new cluster proposals, accepted if ELBO increases, yielding $K+K'$ clusters) and merge moves (combining two clusters if ELBO improves, yielding $K-1$ clusters).
>
> (4) **Gradient**: The DPMM is inferred via variational inference, reducing to calculating the sufficient statistics with the assumption of exponential family, not backpropagation. The scene encoder receives gradients through the KL divergence loss with respect to the fixed cluster parameters, not through categorical samples.
>
> **[W3]**
>
> We acknowledge that the terminology could be more precise. The model does not run a distinct diffusion for all $c$. For each input, the DPMM assigns a single cluster assignment $c$. The diffusion backbone is shared, but $c$ modulates the generation via Adaptive Layer Normalization (AdaLN), analogous to how a Gaussian Mixture Model uses shared component forms with distinct per-component parameters. We will adopt the clearer term "mixture of conditional diffusion experts" in the revision. The dual conditioning on $z$ (continuous scene embedding) and $c$ (discrete cluster assignment) jointly reduces interference: $z$ captures instance-specific features while $c$ routes to scenario-specialized behavior.
>
> **[W4]**
>
> (1) **Position**: Generative replay occurs during training (not inference). When new scenarios arrive, a frozen copy of the old model generates synthetic trajectories for previously learned clusters. These are mixed with new-scenario data during lower part diffusion planner training.
>
> (2) **Memory efficiency**: The memory stores latent $z$ vectors ($d$-dimensional) and cluster parameters not raw multi-modal inputs. At feature dimension $d=16$, NOMAD uses only 0.92MB vs. 3.24GB for raw observation buffers.
>
> (3) **Inference**: At test time, the encoder computes $z$ once, and the DPMM assigns $c$ via a lightweight $O(K)$ operation ($K$ is the active clusters). No replay is involved.
>
>
> **[W5]**
>
> We will include a detailed **algorithm box** in the appendix describing training and inference as outlined above to improve readability.

---

> > ### Author Rebuttal · Reviewer_ruF3 · 2026-04-04
> >
> > The training set-up is very complex, and I think it is concerning it was omitted from the original manuscript, as it raises a lot more other questions, for example: if the diffusion assigns a c, how is the assigment mechanism trained (since this would require some kind of trick to deal with the discrete variable), how did the authors come up with their partial training scheme and can they defend it? Moreover, the setup is still not entirely clear to me, could the authors provide the algorithm they promised in a pdf using an anonymous website? Ideally both for training and evaluation.

---

> > > ### Author Response · Authors · 2026-04-06
> > >
> > > We appreciate the opportunity to clarify these important points in detail.
> > >
> > > **[Q1] About assignment c**
> > >
> > > We would like to respectfully clarify a possible misunderstanding: **the diffusion model does not assign or produce $c$.** The cluster assignment $c$ is determined entirely by the DPMM module in the upper part, using memoVB, not gradient-based optimization. There is no discrete sampling operation in the diffusion model's computational graph, and therefore no need for Gumbel-Softmax or any other gradient trick.
> > >
> > > Concretely, the information flow is:
> > >
> > > 1. **DPMM assigns c** (upper part): Given a scene embedding $z$, the DPMM computes the variational posterior responsibility $r_{nk} = q(v_n = k)$ for each cluster $k$ using sufficient statistics. The assignment $c = \arg\max_k r_{nk}$ is a deterministic lookup after inference.
> > >
> > > 2. **Diffusion model receives $c$ as a fixed input** (lower part): The integer $c$ is mapped through a learnable embedding layer (the "Expert Embedding" block in Fig. 1). This embedding is added to the diffusion timestep embedding, and the combined vector modulates features via Adaptive Layer Normalization (AdaLN), scale and shift operations at each transformer layer.
> > >
> > > This is architecturally identical to how DiT (Peebles & Xie, 2023) conditions on class labels for image generation. In DiT, a class label (integer) is projected to an embedding and injected via AdaLN. Assignment $c$ plays the same role.
> > >
> > > To make this even more concrete: during a single training step, the upper part has already determined that "this scene belongs to cluster 3." The diffusion model then simply looks up the embedding vector for it and uses it to modulate its features. There is no gradient flowing back from the diffusion loss to the cluster assignment. The cluster assignment is as fixed as a dataset label would be.
> > >
> > > **[Q2] Justification for the partial training scheme**
> > >
> > > To clarify the structure: the "alternation" occurs **within the upper part** (scene encoder + DPMM). Specifically:
> > > - **Step A**: Fix the scene encoder, update the DPMM via memoVB using the current embeddings $z$.
> > > - **Step B**: Fix the DPMM cluster parameters, update the scene encoder by minimizing KL divergence between the encoder distribution and the assigned cluster components.
> > > - Steps A and B alternate every fixed number of iterations.
> > > - The lower diffusion planner trains separately on frozen upper outputs ($z$ and $c$), disentangling the training of the upper and lower parts.
> > >
> > > We designed this disentangled training scheme based on three principled motivations:
> > >
> > > (1) **Isolating early-stage noise from lower planner for stable learning.** During the initial training of each new task, the diffusion planner generates significant gradient noise. If the encoder and planner were trained entangled (with an added loss coupling them), planner gradients would corrupt the learned feature representations.
> > >
> > > (2) **Enforcing knowledge-preserving and trajectory-planning role separation.** The upper part learns scenario-level representations and discovers cluster structure, while the lower part generates trajectories. Having each component strictly adhere to its role avoids conflicting optimization objectives.
> > >
> > > (3) **Protecting non-parametric memory for continual learning.** If the encoder's feature space fluctuates from entangled planner gradients, the DPMM cannot perform stable clustering (birth or merge). Decoupling ensures DPMM operates on a stable embedding distribution, preserving cluster centers for previously learned knowledge, as shown in the **ablation study** in the original paper and the following experiments.
> > >
> > > **Further Empirical validation**:
> > >
> > > | Method | interPlan | Val14(R) | Val14(NR) | Test14-Hard(R) | Test14-Hard(NR) |
> > > |-|-|-|-|-|-|
> > > | NOMAD (disentangled) | **70** | **95** | **94** | **85** | **78** |
> > > | Entangled | 66 | 93 | 91 | 82 | 76 |
> > >
> > > The disentangled training scheme outperforms the entangled variant (scene encoder and diffusion planner are trained with an added loss; DPMM inferred by memoVB), by 4 points on interPlan and 3 points on Test14-Hard(R).
> > >
> > > The training loss curve (anonymous [link](https://anonymous.4open.science/r/NOMAD_Rebuttal/loss.png)) shows smoother convergence for the disentangled training scheme (NOMAD, blue curve), while entangled training exhibits oscillations consistent and higher convergence loss with the gradient interference described above (Entangled, red dash curve).
> > >
> > > **[Q3] Algorithm in PDF**
> > >
> > > As promised, we have prepared a detailed algorithm PDF covering both training and inference ([link](https://anonymous.4open.science/r/NOMAD_Rebuttal/NOMAD_Algorithm.pdf)):
> > >
> > > It contains:
> > > - **Algorithm 1**: Training (upper alternating optimization + lower diffusion planner training with replay)
> > > - **Algorithm 2**: Inference (encoding → DPMM assignment → conditional diffusion generation → classifier guidance)
> > >
> > > We hope these clarifications fully resolve the remaining concerns. We are happy to answer any further questions.

---

### Decision · Program_Chairs · 2026-04-30

**Decision:**

Accept (regular)

**Comment:**

NOMAD integrates a Dirichlet Process Mixture Model with diffusion-based trajectory generation for lifelong trajectory planning, featuring generative replay to mitigate catastrophic forgetting. All three reviewers give Weak Accept, recognizing the elegant architecture, strong nuPlan/interPlan results, and practical forgetting avoidance. The rebuttal substantially improved clarity, adding t-SNE cluster visualizations (8 semantically distinct clusters with 100% purity) and encoder robustness analysis. Remaining concerns include whether the single conditioned diffusion model truly constitutes a "mixture of experts," unaddressed memory growth for long-term deployment, and limited validation beyond simulation. These are, in the perspective of AC, beyond the scope of this work; the core contribution is solid.